# Boosting World Models Learning via Latent-Space Value Alignment

**Xingyu Jiang** [1]  **Yuheng Pan** [1]  **Mukang You** [1]  **Xiuhui Zhang** [2]  **Ning Gao** [1]  **Guanwei Yan** [3]  **Hao Li** [3]  **Yue Deng** [2 4]

## Abstract

Model-based reinforcement learning aims to construct world models for efficient sampling. Current mainstream algorithms can be broadly categorized into two paradigms: maximum likelihood and value-aware world models. The former employs structured Recurrent/Transformer State-Space Models to capture environmental dynamics but overlooks task-relevant features. The latter prioritizes decision-critical states but suffers from sub-optimal performance. While recent efforts have sought to integrate these approaches, they typically rely on auxiliary modules or heavy external priors that significantly increase computational complexity. In this work, we propose a Value-Aligned World Model, a minimalist framework designed to synergize these two paradigms with negligible overhead. Specifically, We introduce an intrinsic latent-space value-alignment regularization that compels the world model to prioritize task-relevant features while maintaining the structural integrity of stochastic dynamics. To ensure stable optimization, we develop an adaptive weighting mechanism that acts as a self-regulating curriculum, balancing reconstruction fidelity with decision-making utility. Extensive experiments on Atari 100k and DeepMind Control benchmarks demonstrate that our algorithm consistently boosts existing methods with minimal added code and computational overhead. Code is available at supplementary material.

## 1. Introduction

Driven by trial-and-error mechanisms, deep reinforcement learning (DRL) has achieved remarkable progress across

[1]School of Astronautics, Beihang University, Beijing, China [2]School of Artificial Intelligence, Beihang University, Beijing, China [3]Chengdu Aviation Corporation, Chengdu, China [4]Beijing Zhongguancun Academy, Beijing, China. Correspondence to: Yue Deng <ydeng@buaa.edu.cn>.

*Proceedings of the $43^{rd}$ International Conference on Machine Learning*, Seoul, South Korea. PMLR 306, 2026. Copyright 2026 by the author(s).

diverse domains, including game play (Vinyals et al., 2019; Berner et al., 2019), robotic control (Ju et al., 2022; Jiang et al.), and large model fine-tuning (Guo et al., 2025; Zheng et al., 2025). Despite these successes, the prohibitive sample complexity of DRL often hampers its deployment in real-world applications. To mitigate this, model-based reinforcement learning (MBRL) has gained significant traction as a promising paradigm. By learning a world model to simulate environment dynamics, MBRL reduces the reliance on frequent real-world interactions and facilitates highly efficient sampling. According to their training objectives, current MBRL algorithms are broadly categorized into two paradigms: maximum likelihood(Hafner et al., 2019a; 2020; 2023; Ma et al., 2023; Zhang et al., 2023; Burchi & Timofte, 2025; Zhang et al., 2025a) and value-aware world models(Farahmand et al., 2017; Voelcker et al., 2022; 2025). The former utilizes variational inference to model environment dynamics via structured Recurrent/Transformer State-Space Models (RSSM/TSSM), yet it often prioritizes holistic reconstruction indiscriminately, thereby overlooking the varying importance of different representations. Conversely, the latter incorporates value functions to emphasize task-relevant features but frequently suffers from sub-optimal empirical performance. Consequently, maximum likelihood remains the dominant paradigm, while the advancement of value-aware world models has been comparatively limited.

However, from a value-aware perspective, current maximum likelihood-based world models suffer from two core limitations: **(1) Interference from task-irrelevant distractors.** While maximum likelihood objectives prioritize the high-fidelity reconstruction of all observations, much of this sensory data remains decoupled from task rewards in complex environments. The resulting indiscriminate treatment of all feature dimensions often exhausts modeling capacity on decision-irrelevant noise, thereby diluting task-critical representations and inducing significant inefficiencies in complex or high-redundancy settings. **(2) Objective mismatch between modeling and behavior.** Since likelihood-based losses prioritize statistical divergence over decision-making significance, a high-fidelity world model does not necessarily yield an optimal policy, as even statistically negligible prediction errors can trigger catastrophic behavioral failures. Lacking the guidance of a value function, such models fail to distinguish decision-critical dynamics from secondary

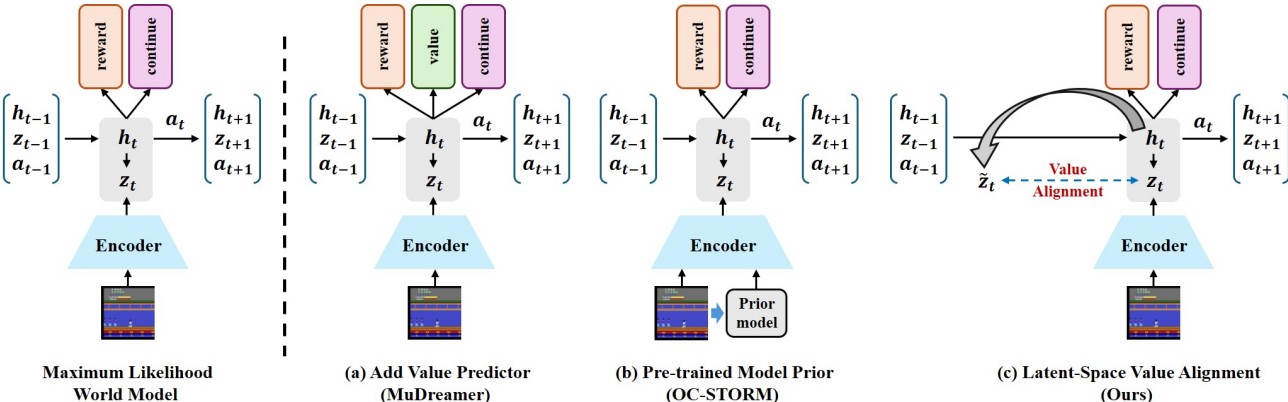

*Figure 1.* Value-aware paradigms: (a) explicit value prediction, (b) external pre-trained priors, and (c) latent-space value alignment.

details, thereby preventing the prioritized optimization of representations essential for successful behavior.

To address these limitations, recent research has sought to incorporate value-aware mechanisms within the maximum likelihood framework. As illustrated in Fig.1(a), methods represented by MuDreamer(Burchi & Timofte, 2024) employ an explicit prediction strategy, performing joint optimization by incorporating an auxiliary value branch into the RSSM. Although explicit value prediction can enforce value awareness in the posterior, this approach often suffers from gradient dominance, where high-dimensional reconstruction gradients overwhelm sparse value signals. More importantly, even if the posterior captures value-relevant features, the standard dynamics loss does not guarantee their propagation to the prior, resulting in a prior-posterior value awareness gap. Conversely, as shown in Fig.1(b), works such as OC-STORM(Zhang et al., 2026) pivot towards an external augmentation approach, leveraging pre-trained models (e.g., object detectors) as priors to enhance the discriminative power of representations. While this paradigm effectively suppresses task-irrelevant background noise, its reliance on heavy external models introduces significant computational and memory overheads, while also complicating the training pipeline. In stark contrast, as depicted in Fig.1(c), we propose a minimalist Latent-Space Value Alignment mechanism. Discarding redundant auxiliary branches and heavy external dependencies, we impose intrinsic value constraints directly between the prior and posterior distributions of state transitions. This mechanism seamlessly injects value guidance into dynamics modeling, achieving a deep synergy between task-aware representations and stochastic dynamics with minimal modifications.

**Our Contribution.** In this paper, we address two fundamental challenges inherent in maximum likelihood world models: interference from task-irrelevant distractors and the objective mismatch between modeling and behavior. To this end, we introduce the Value-aligned World Model, a novel framework designed to bridge the gap between world model learning and decision-making through value alignment. At

its core is a Latent-Space Value-Alignment Regularization (Var) term, which imposes intrinsic constraints on the latent state distributions, compelling the world model to prioritize decision-critical features while simultaneously capturing complex environmental dynamics. To ensure stable optimization, we develop an adaptive weighting mechanism paired with a warm-up phase to dynamically balance the trade-off between reconstruction fidelity and value alignment. We demonstrate the efficacy of our approach by integrating it into state-of-the-art baselines, DreamerV3 (Hafner et al., 2023) and STORM (Zhang et al., 2023), across the Atari 100k and DeepMind Control benchmarks. Extensive experiments show that our method consistently yields significant performance gains; notably, on Atari 100k, it improves DreamerV3's human-normalized mean score from 1.10 to 1.34 and the median from 0.58 to 1.00. These results underscore the robustness of our framework, confirming that value alignment provides consistent enhancements across a wide spectrum of environments rather than relying on outlier improvements. Meanwhile, our algorithm is better regraded as a plug-and-play module that necessitates only minimal modifications and introduces negligible computational overhead when integrated into existing MBRL frameworks.[2]

## 2. Preliminaries

**Reinforcement Learning:** We formalize the environment as a Markov Decision Process(Puterman, 1990) defined as a tuple $(S, A, r(s, a), P(s'|s, a), \gamma)$, where $S$ and $A$ represent the state and action spaces, $r(s, a)$ is the reward function, $P(s'|s, a)$ denotes the state transition dynamics and $\gamma \in (0, 1)$ is the discount factor. The objective of reinforcement learning is to optimize the cumulative reward over time.

**Model-based Reinforcement Learning:** MBRL leverages a latent world model to approximate environment dynamics $P(z'|z, a)$, where $z$ represents the latent state encoded from observations. We focus on the learning through imagination

---

[2]Due to space constraints, a comprehensive discussion of Related Work Section is deferred to Appendix B.

paradigm, which iterates through three phases: experience collection, world model learning, and policy optimization. Crucially, real-world interactions are reserved solely for training the world model, whereas the agent's policy is optimized entirely within the synthetic trajectories.

## 3. Methods

In this section, we first establish the empirical motivation for synergizing maximum likelihood and value-aware world models, highlighting how they mutually reinforce each other. Subsequently, taking DreamerV3(Hafner et al., 2023) as a representative backbone, we demonstrate the integration of the value-alignment regularization term into maximum likelihood world model optimization objective.

### 3.1. Maximum Likelihood and Value-Aware Paradigms

To facilitate a seamless integration of maximum likelihood and value-aware paradigms, it is essential to first investigate their respective intrinsic mechanisms and complementary properties. Maximum likelihood world models, powered by RSSM/TSSM architectures, excel at capturing stochastic dynamics through structured variational inference. However, their reliance on reconstruction-based objectives often lacks feature discrimination, causing the model to squander its capacity on task-irrelevant background noise. This indiscriminate modeling leads to a fundamental task misalignment between reconstruction fidelity and decision-making utility, particularly under limited model capacity or high informational redundancy. In contrast, value-aware world models prioritize decision-critical features, yet they are often constrained by simplistic RNN backbones that struggle with complex environmental transitions. Furthermore, reliance on sparse value signals, compounded by the non-convexity of objectives like VAML (Voelcker et al., 2022), often triggers dynamics collapse and training instabilities.

Consequently, bridging these two paradigms offers a natural progression toward harmonizing modeling stability with task-specific focus. By leveraging the robust RSSM/TSSM backbone alongside value-aware attention filtering, we rectify the statistical biases of pure likelihood-based methods,

focusing representation resources on decision-critical features to achieve superior decision effectiveness while preserving dynamical integrity.

As illustrated in Fig.2, we conducted a series of experiments to empirically validate the aforementioned analysis. Specifically, we evaluate the maximum likelihood baselines, DreamerV3(Hafner et al., 2023) and STORM(Zhang et al., 2023), alongside the value-aware algorithm, VaGraM (Voelcker et al., 2022), on the visual games Gopher and Krull, which respectively evaluate the short-term and long-term planning capabilities. Fig.2(a) presents results across different world model capacities (input size: $64\times64$), with three settings: small (1M), medium (12M) and large (25M). The results indicate that when modeling capacity is limited, the maximum likelihood objective struggles to capture essential dynamics, leading to significant performance drops; in such scenarios, value-alignment regularization redirects the model's focus toward decision-critical states, yielding substantial improvements. Fig.2(b) presents the results with different image input sizes (world model capacity: 12M), using three configurations: $64\times64$, $96\times96$ and $128\times128$. As the resolution increases, the influx of task-irrelevant distractors overwhelms the maximum likelihood loss, hindering its ability to extract task-relevant features and causing performance degradation. Notably, the introduction of value-alignment regularization significantly alleviates this issue by ensuring the latent space remains anchored to behavioral utility despite increased sensory redundancy.

These experiments demonstrate that for standard maximum likelihood algorithms (DreamerV3 and STORM), incorporating value-alignment regularization yields significant benefits, particularly when model capacity is constrained or input information is highly redundant. In contrast, the pure value-aware baseline, VaGraM, consistently underperforms compared to maximum likelihood approaches across all settings. This observation reinforces a critical insight for MBRL: a robust world model architecture (RSSM/TSSM) is essential to establish a stable performance foundation, whereas value-alignment awareness serves to unlock superior decision-making capabilities, especially in challenging deployment scenarios.

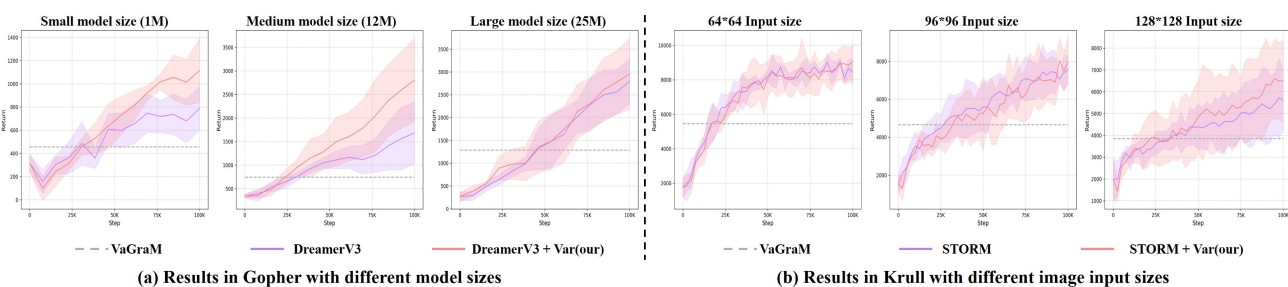

**(a) Results in Gopher with different model sizes**      **(b) Results in Krull with different image input sizes**

*Figure 2.* Experimental results across different model sizes and input dimensions.

## 3.2. Value-aligned World Model Learning

Building upon the insights from previous section regarding the necessity of a robust structural backbone, we implement the world model using a Recurrent State-Space Model, following the architectural principles of DreamerV3 (Hafner et al., 2023). Parameterized by $\alpha$, this model is designed to capture complex environmental dynamics and synthesize compact state representations, thereby enabling the coherent imagination of future trajectories. Specifically, given an image observation $o_t$, we map it to a latent stochastic representation $z_t$ via an encoder network, which is a VAE with categorical latents. A temporal sequence model then predicts the next recurrent state $h_t$ based on the previous recurrent state $h_{t-1}$, latent representation $z_{t-1}$, and action $a_{t-1}$. Finally, the model state $s_t = \{h_t, z_t\}$, formed by concatenating $h_t$ and $z_t$, is used to predict the environment reward $r_t$, the episode continuation flag $c_t$, and to reconstruct the input $o_t$ via a decoder network. Specifically, the encoder and decoder use convolutional neural networks (CNNs) for image inputs and multilayer perceptrons (MLPs) for vector inputs. The sequence model is based on a recurrent neural networks (RNNs), while the dynamics, reward, and continuation predictors are implemented as MLPs. The components of the RSSM-based world model are illustrated below:

$$
\begin{aligned}
\text{Sequence model:} \quad & h_t = f_\alpha(h_{t-1}, z_{t-1}, a_{t-1}) \\
\text{Encoder Network:} \quad & z_t \sim q_\alpha(z_t | h_t, o_t) \\
\text{Dynamics Predictor:} \quad & \tilde{z}_t \sim p_\alpha(\tilde{z}_t | h_t) \\
\text{Reward Predictor:} \quad & \tilde{r}_t \sim p_\alpha(\tilde{r}_t | s_t) \\
\text{Continue Predictor:} \quad & \tilde{c}_t \sim p_\alpha(\tilde{c}_t | s_t) \\
\text{Decoder Network:} \quad & \tilde{o}_t \sim p_\alpha(\tilde{o}_t | s_t)
\end{aligned}
\tag{1}
$$

**World Model Loss Function:** Given a batch size $B$ and sequence length $T$, with input observations $o_{1:T}$, actions $a_{1:T}$, rewards $r_{1:T}$, and episode continuation flags $c_{1:T}$, the world model is optimized by minimizing the following loss:

$$
L_{world} = \frac{1}{B \times T} \sum_{b=1}^{B} \sum_{t=1}^{T} \Big[ L_{pred} + L_{dyn} + \beta_{var} L_{var} \Big]
\tag{2}
$$

The prediction loss $L_{\text{pred}}$ is computed using symlog squared loss to train the decoder network and reward predictor, while logistic regression is applied to train the continuation predictor. The dynamics loss $L_{\text{dyn}}$ is used to train the sequence model by minimizing the KL divergence between the predicted distribution $p_\alpha(\tilde{z}_t | h_t)$ and the next encoder representation $q_\alpha(z_t | h_t, o_t)$. In practice, $L_{\text{dyn}}$ utilizes the stop gradient operator $sg(\cdot)$ to prevent backpropagation of gradients. Additionally, a representation loss is introduced to

encourage the encoder to learn more predictable state representations. These two loss components: the prediction loss and dynamics loss, form the standard world model loss function, as follows, with $\beta_{\text{dyn}} = 0.5$ and $\beta_{\text{rep}} = 0.1$:

$$
\begin{aligned}
L_{pred} = & -\ln p_\alpha(o_t | s_t) - \ln p_\alpha(r_t | s_t) - \ln p_\alpha(c_t | s_t) \\
L_{dyn} = & \beta_{dyn} \text{KL} \big[ \text{sg}(q_\alpha(z_t | h_t, o_t)) \, || \quad p_\alpha(\tilde{z}_t | h_t) \big] \\
& + \beta_{reg} \text{KL} \big[ \quad q_\alpha(z_t | h_t, o_t) \, || \, \text{sg}(p_\alpha(\tilde{z}_t | h_t)) \big]
\end{aligned}
\tag{3}
$$

The Value-Alignment Regularization $L_{\text{var}}$ serves as a critical constraint to bridge the objective mismatch between reconstruction and decision-making. Unlike traditional value-aware paradigms that explicitly distort dynamics learning via one-step value prediction errors or unstable value-gradient weighting(Voelcker et al., 2022), we implicitly inject value awareness through latent-space alignment. Specifically, we leverage the distributional output of the value network $V_\theta$ (parameterized as a categorical distribution in DreamerV3) rather than the final scalar estimate. By minimizing the KL divergence between the value distributions inferred from the prior and posterior state representations, we enforce a high-level semantic consistency that promotes the capture of task-relevant features. Structurally, we follow the same design as the dynamics loss $L_{\text{dyn}}$, introducing the stop gradient operator $sg(\cdot)$ to stabilize the training process. The specific formulation is as follows, where $\tilde{s}_t = \{h_t, \tilde{z}_t\}$:

$$
\begin{aligned}
L_{var} = & \beta_{dyn} \text{KL} \big[ \text{sg}(V_\theta(v_t | s_t)) \, || \quad V_\theta(\tilde{v}_t | \tilde{s}_t) \big] \\
& + \beta_{reg} \text{KL} \big[ \quad V_\theta(v_t | s_t) \, || \, \text{sg}(V_\theta(\tilde{v}_t | \tilde{s}_t)) \big]
\end{aligned}
\tag{4}
$$

To ensure stable optimization and manage the trade-off between value alignment and dynamics learning, we introduce an adaptive weighting mechanism paired with a warm-up phase. This is implemented through an indicator function $\mathbb{1}_{t > 10^4}$ and a dynamic coefficient $\beta_{\text{var}}$. Specifically, the first 10,000 training steps are designated as a warm-up phase, during which value-alignment regularization is deactivated. This prevents the world model from being misled by the highly stochastic and unreliable value estimations typical of early-stage exploration. Beyond the warm-up phase, the core challenge lies in balancing task-aware alignment with the fundamental dynamics loss $L_{\text{dyn}}$. We posit that value alignment is most effective once the world model has learned reliable environmental dynamics, while premature or excessive alignment may distort representation learning. To this end, we use the inverse dynamics loss $L_{\text{dyn}}$ as the adaptive weight $\beta_{\text{var}}$, forming a self-regulating curriculum. When $L_{\text{dyn}}$ is high, value alignment is suppressed to prioritize structural dynamics modeling; as $L_{\text{dyn}}$ decreases, $\beta_{\text{var}}$ increases, shifting optimization toward decision-critical,

value-sensitive features. This hierarchical strategy enables value alignment to improve performance only after a stable dynamical foundation has been established.

$$\beta_{var} = 1/\max\big(1, \mathrm{sg}\big(\mathrm{KL}\big[q_\alpha(z_t|h_t,o_t)||p_\alpha(\tilde{z}_t|h_t)\big]\big)\big) \tag{5}$$

### 3.3. Agent behavior learning

Following DreamerV3(Hafner et al., 2023), the actor and critic are trained on imagined trajectories generated by the world model. We freeze the world model and unroll latent trajectories to estimate state values and optimize the policy: the actor maximizes expected return, while the critic regresses value targets derived from imagined rewards. To fairly attribute performance gains to our value-aligned world model, we keep the behavior learning objectives, hyperparameters, and architectural configurations identical to the standard DreamerV3 baseline. Detailed implementation specifications are provided in Appendix A.

## 4. Experiments

### 4.1. Benchmarks and Baselines

**(1) The Atari 100k benchmark** (Kaiser et al., 2019) consists of 26 Atari games with discrete action controls, utilizing a budget of 400k environment frames, equivalent to approximately two hours of actual gameplay. In practice,

we choose VaGraM (Voelcker et al., 2022), MuDreamer (Burchi & Timofte, 2024), DreamerV3 (2023 v1) (Hafner et al., 2023), DreamerV3 (2025 v2) (Hafner et al., 2025), STORM (Zhang et al., 2023) and OC-STORM (Zhang et al., 2026), as baselines for comparison.

**(2) The DeepMind Control Suite** (Tassa et al., 2018) is divided into two components based on input types. The Proprio Control part consists of 18 continuous action tasks with proprioceptive vector inputs, using a budget of 500K environment steps. The Visual Control part comprises 20 continuous control tasks and a budget of 1M environment steps. In practice, we choose TD-MPC2 (Hansen et al., 2023), MuDreamer (Burchi & Timofte, 2024), TWISTER (Burchi & Timofte, 2025), DMPO (Abdolmaleki et al., 2018), D4PG (Barth-Maron et al., 2018), DreamerV3 (2023 v1) (Hafner et al., 2023) and DreamerV3 (2025 v2) (Hafner et al., 2025) as baselines for comparison.

**Note on Baseline Selection:** It is important to distinguish between the two iterations of DreamerV3: the initial v1 release (2023) and the updated v2 version (2025). The v1 architecture operates as a pure maximum likelihood world model. In contrast, the v2 version not only scales up model capacity but also incorporates value awareness via a "replay value loss," which resembles the explicit prediction strategy illustrated in Fig.1(a). To strictly validate the efficacy of our Latent-Space Value Alignment without confounding factors from pre-existing value-aware mechanisms, we conduct our experiments based on the 2023 v1 implementation.

*Table 1.* Quantitative results on the Atari 100k benchmark. We show average scores over 5 seeds.

| Game | Random | Human | VaGraM (2022) | MuDreamer (2024) | DreamerV3 (2025 v2) | DreamerV3 (2023 v1) | DreamerV3 + Var (Ours) | OC-STORM (2025) | STORM (2023) | STORM + Var (Ours) |
|---|---|---|---|---|---|---|---|---|---|---|
| Alien | 227.8 | 7127.7 | 647.8 | 951.0 | 1118.0 | 875.88 | 1233.2(↑40.8%) | 1101.4 | 1054.3 | 1361.4(↑29.1%) |
| Amidar | 5.8 | 1719.5 | 82.9 | 153.0 | 97.0 | 143.7 | 185.4(↑29.0%) | 162.7 | 177.29 | 248.36(↑40.1%) |
| Assault | 222.4 | 742.0 | 662.4 | 891.0 | 683.0 | 843.7 | 981.38(↑16.3%) | 1270.4 | 715.9 | 752.55(↑5.1%) |
| Asterix | 210.0 | 8503.3 | 924.8 | 1411.0 | 1062.0 | 1102.5 | 1162.6(↑5.5%) | 1753.5 | 1276.0 | 1535.0(↑20.3%) |
| BankHeist | 14.2 | 753.1 | 127.0 | 156.0 | 398.0 | 1072.0 | 1121.2(↑4.6%) | 1075.2 | 1060.5 | 935.0(↓11.8%) |
| BattleZone | 2360.0 | 37187.7 | 4145.0 | 12080.0 | 20300.0 | 11138.0 | 12750.0(↑14.5%) | 4590.0 | 7080.0 | 10140.0(↑43.2%) |
| Boxing | 0.1 | 12.1 | 64.7 | 96.0 | 82.0 | 80.3 | 87.4(↑8.9%) | 92.2 | 78.6 | 83.0(↑5.6%) |
| Breakout | 1.7 | 30.5 | 12.8 | 34.0 | 10.0 | 25.3 | 45.6(↑79.9%) | 52.5 | 20.88 | 26.43(↑26.6%) |
| ChopperCommand | 811.0 | 7387.8 | 956.3 | 808.0 | 2222.0 | 1438.0 | 1826.0(↑27.0%) | 2090.0 | 1768.0 | 1695.0(↓4.1%) |
| CrazyClimber | 10780.5 | 35829.4 | 43215.6 | 96128.0 | 86225.0 | 89900.0 | 81720.0(↓9.1%) | 84111.0 | 47473.0 | 57335.0(↑20.8%) |
| DemonAttack | 152.1 | 1971.0 | 182.4 | 553.0 | 577.0 | 223.9 | 227.2(↑1.5%) | 411.3 | 194.6 | 204.6(↑5.1%) |
| Freeway | 0.0 | 29.6 | 18.2 | 5.0 | 0.0 | 30.2 | 31.6(↑4.6%) | 0.0 | 29.7 | 32.0(↑7.8%) |
| Frostbite | 65.2 | 4334.7 | 258.4 | 1652.0 | 3377.0 | 1628.0 | 347.9(↓78.6%) | 259.6 | 258.8 | 260.2(↑0.5%) |
| Gopher | 257.6 | 2412.5 | 772.7 | 1500.0 | 2160.0 | 1683.9 | 2807.0(↑66.7%) | 4456.8 | 8551.0 | 13509.6(↑58.0%) |
| Hero | 1027.0 | 30826.4 | 2147.9 | 8272.0 | 13354.0 | 4994.4 | 9360.6(↑87.4%) | 6441.4 | 12249.2 | 12574.0(↑2.7%) |
| Jamesbond | 29.0 | 302.8 | 246.5 | 409.0 | 540.0 | 332.0 | 542.0(↑63.3%) | 347.0 | 446.4 | 462.5(↑3.6%) |
| Kangaroo | 52.0 | 3035.0 | 824.2 | 4380.0 | 2643.0 | 1529.2 | 3650.4(↑138.7%) | 4218.0 | 1542.0 | 3322.6(↑115.4%) |
| Krull | 1598.0 | 2665.5 | 5685.8 | 9644.0 | 8171.0 | 8364.8 | 9821.4(↑17.4%) | 9714.6 | 8360.1 | 8896.5(↑6.4%) |
| KungFuMaster | 258.5 | 22736.3 | 13142.0 | 26832.0 | 25900.0 | 16375.0 | 21075.0(↑28.7%) | 24988.0 | 15760 | 26615.0(↑68.9%) |
| MsPacman | 307.3 | 6951.6 | 1248.6 | 2311.0 | 1521.0 | 1947.0 | 1749.5(↓10.1%) | 2400.7 | 1906.9 | 2417.3(↑26.8%) |
| Pong | -20.7 | 14.6 | 13.4 | 18.0 | -4.0 | 19.1 | 19.8(↑4.0%) | 20.6 | 20.6 | 20.2(↓1.9%) |
| PrivateEye | 24.9 | 69571.3 | 114.7 | 1042.0 | 3238.0 | 2331.2 | -115.6(↓104.9%) | 85.0 | 414.4 | 2584.7(↑523%) |
| Qbert | 163.9 | 13455 | 864.2 | 4061.0 | 2921.0 | 1223.5 | 2267.8(↑85.4%) | 4546.2 | 2912.5 | 4243.4(↑45.7%) |
| RoadRunner | 11.5 | 7845.0 | 6216.8 | 8460.0 | 19230.0 | 9868.6 | 14704.0(↑49.0%) | 20482.0 | 11523.0 | 13999.0(↑21.5%) |
| Seaquest | 68.4 | 42054.7 | 412.5 | 428.0 | 962.0 | 513.2 | 546.3(↑6.5%) | 712.2 | 441.4 | 430.0(↓2.6%) |
| UpNDown | 533.4 | 11693.2 | 5845.2 | 26494.0 | 46910.0 | 12679.2 | 18485.4(↑45.8%) | 6623.2 | 6406.4 | 8982.6(↑40.2%) |
| Superhuman (↑) | 0 | N/A | 3 | 11 | 7 | 10 | 13(↑3) | 12 | 9 | 12(↑3) |
| Mean (↑) | 0.00 | 1.00 | 0.62 | 1.26 | 1.25 | 1.10 | 1.34(↑0.24) | 1.35 | 1.14 | 1.36(↑0.22) |
| Median (↑) | 0.00 | 1.00 | 0.20 | 0.43 | 0.49 | 0.58 | 1.00(↑0.42) | 0.44 | 0.51 | 0.81(↑0.30) |

## 4.2. Results on Atari 100k

Tab.1 presents the quantitative results of applying value-alignment regularization to DreamerV3 (2023 v1) (Hafner et al., 2023) and STORM(Zhang et al., 2023) on the Atari 100k benchmark, while Fig.11 shows the training curves. To ensure fair comparison, we retrained both DreamerV3 and STORM using identical hyperparameters. Following previous work, we used human-normalized metrics to evaluate performance across 26 games, comparing mean and median scores. The results demonstrate consistent performance improvements: for DreamerV3, 24 out of 26 games showed improvements, with the average score increasing from 1.10 to 1.34 and the median from 0.58 to 1.00. Similarly, STORM improved in 24 games, with the average score rising from 1.14 to 1.36 and the median from 0.51 to 0.81. Notably, games such as KungFuMaster, Gopher, Qbert and Kangaroo, where small target characters are crucial, exhibited particularly significant performance gains.

Furthermore, our method demonstrates superior robustness compared to recent value-aware approaches. In contrast to MuDreamer (Burchi & Timofte, 2024), which employs an explicit value prediction branch, our DreamerV3 + Var achieves a significantly higher median score (1.00 vs. 0.43) and a superior mean (1.34 vs. 1.26). This indicates that our implicit latent alignment effectively mitigates the optimization instability often associated with explicit value objectives. Meanwhile, compared to OC-STORM (Zhang et al., 2026), which relies on external object detectors as priors, our STORM + Var achieves a comparable mean (1.36 vs. 1.35) but nearly doubles the median score (0.81 vs. 0.44). This result highlights the efficiency of our approach: we achieve better performance and stability across diverse environments without additional computational burden.

Despite these broad gains, we observe performance regres-

sions in specific environments such as PrivateEye and Frostbite. We hypothesize that this stems from the extreme reward sparsity inherent in these tasks, which demand extensive initial exploration. In such scenarios, value targets remain predominantly noisy and non-informative throughout much of the training process. Forcing the world model to align with these unreliable signals may induce a form of "value-driven overfitting", which suppresses the latent-space variance essential for effective exploration. This suggests a potential trade-off between task-specific focus and exploratory breadth in high-sparsity domains.

## 4.3. Results on DeepMind Control Suite

Tab.2 presents the quantitative results of applying value-alignment regularization to DreamerV3 (2023 v1) (Hafner et al., 2023) on the DMC Suite benchmark. To ensure a fair comparison, DreamerV3 was retrained with identical hyperparameters for both input modalities. The results show consistent performance improvements across continuous control tasks: with visual inputs, 15 out of 20 tasks saw improvements, with the average score increasing from 792 to 827 and the median from 877 to 894; with vector inputs, performance improved in 13 out of 18 tasks, with the average score rising from 805 to 817 and the median from 881 to 901. Fig.12 shows the training curves for the DMC Suite benchmark. These results demonstrate that our approach accelerates the convergence of MBRL algorithms, especially in tasks like Pendulum Swingup and Walker Walk.

Notably, the performance gains on the DMC benchmark, while positive, appear more moderate compared to the substantial improvements observed in the Atari 100k tasks. We argue that this discrepancy actually validates the underlying mechanism of our proposed value-alignment regularization. In the Atari domain, environments are characterized by high-dimensional visual inputs with significant

*Table 2.* Quantitative results on the DMC suite benchmark. We show average scores over 5 seeds.

| Task | TD-MPC2 (2023) | MuDreamer (2024) | TWISTER (2025) | DreamerV3 (2025 v2) | DreamerV3 (2023 v1) | DreamerV3+Var (Ours) | DMPO (2018) | D4PG (2018) | DreamerV3 (2025 v2) | DreamerV3 (2023 v1) | DreamerV3+Var (Ours) |
|---|---|---|---|---|---|---|---|---|---|---|---|
| Input types | | | | Visual Image Inputs | | | | | Proprioceptive Inputs | | |
| Environment steps | 1M | 1M | 1M | 1M | 1M | 1M | 500K | 500K | 500K | 500K | 500K |
| Acrobot Swingup | 216 | 304.6 | 239 | 229 | 314 | 367(↑16.7%) | 103 | 124 | 154.5 | 261 | 295(↑13.1%) |
| Ball In Cup Catch | 717 | 930.8 | 967 | 972 | 953 | 967(↑1.5%) | 968 | 968 | 958.2 | 968 | 965(↓0.4%) |
| Cartpole Balance | 931 | 990.4 | 998 | 993 | 998 | 999(↑0.08%) | 999 | 999 | 990.5 | 997 | 999(↑0.2%) |
| Cartpole Balance Sparse | 1000 | 1000.0 | 1000 | 964 | 1000 | 1000(0.00%) | 999 | 974 | 996.8 | 989 | 990(↑0.1%) |
| Cartpole Swingup | 808 | 823.0 | 819 | 861 | 866 | 865(↓0.2%) | 860 | 875 | 850.0 | 872 | 865(↓0.7%) |
| Cartpole Swingup Sparse | 739 | 582.0 | 735 | 759 | 520 | 756(↑45.6%) | 438 | 752 | 468.1 | 802 | 800(↓0.3%) |
| Cheetah Run | 550 | 872.5 | 694 | 836 | 917 | 916(↓0.1%) | 650 | 624 | 575.9 | 748 | 834(↑11.4%) |
| Finger Spin | 986 | 603.6 | 976 | 589 | 520 | 602(↑15.7%) | 769 | 823 | 937.2 | 536 | 537(↑0.2%) |
| Finger Turn Easy | 789 | 915.4 | 924 | 878 | 888 | 914(↑3.0%) | 620 | 612 | 745.4 | 889 | 892(↑0.3%) |
| Finger Turn Hard | 872 | 886.5 | 910 | 904 | 895 | 885(↓1.0%) | 495 | 421 | 841.0 | 975 | 977(↑0.2%) |
| Hopper Hop | 211 | 311.8 | 314 | 227 | 325 | 336(↑3.3%) | 68 | 80 | 111.0 | 236 | 238(↑1.0%) |
| Hopper Stand | 803 | 883.9 | 932 | 903 | 938 | 933(↓0.5%) | 549 | 762 | 573.2 | 862 | 910(↑5.5%) |
| Pendulum Swingup | 743 | 806.7 | 832 | 744 | 807 | 812(↑0.6%) | 834 | 759 | 766.0 | 805 | 852(↑5.8%) |
| Quadruped Run | 362 | 627.8 | 652 | 617 | 782 | 824(↑5.4%) | - | - | - | - | - |
| Quadruped Walk | 253 | 860.0 | 905 | 811 | 810 | 902(↑11.4%) | - | - | - | - | - |
| Reacher Easy | 971 | 907.0 | 933 | 951 | 924 | 961(↑4.0%) | 961 | 960 | 947.1 | 962 | 969(↑0.7%) |
| Reacher Hard | 877 | 733.0 | 566 | 862 | 759 | 797(↑4.9%) | 968 | 937 | 936.2 | 965 | 960(↓0.5%) |
| Walker Run | 728 | 740.9 | 711 | 684 | 688 | 764(↑11.0%) | 493 | 616 | 632.7 | 726 | 716(↓1.4%) |
| Walker Stand | 916 | 964.3 | 977 | 976 | 983 | 985(↑0.2%) | 975 | 947 | 956.9 | 967 | 976(↑0.9%) |
| Walker Walk | 945 | 949.8 | 951 | 961 | 960 | 961(↑0.2%) | 942 | 969 | 935.7 | 930 | 933(↑0.3%) |
| Task mean | 721 | 784.7 | 802 | 786 | 792 | 827(↑4.4%) | 705 | 733 | 743.1 | 805 | 817(↑1.5%) |
| Task median | 796 | 849.6 | 908 | 861 | 877 | 894(↑1.9%) | 801 | 792 | 845.5 | 881 | 901(↑2.3%) |

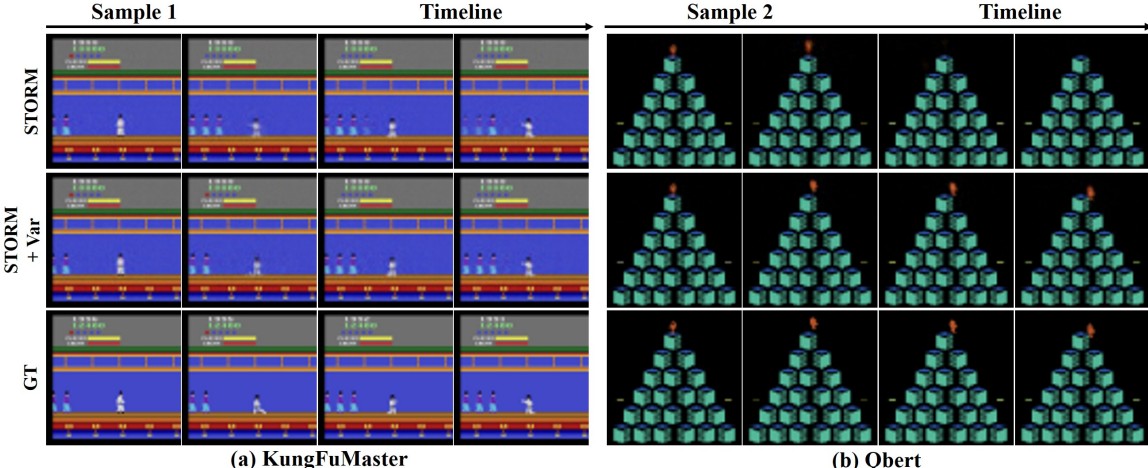

*Figure 3.* Imagined trajectories from the world model in KungFuMaster and Qbert games

task-irrelevant distractors and small, critical objects. In contrast, DMC tasks typically feature proprioceptive inputs or clean visual observations where most state dimensions are inherently task-relevant. In these high signal-to-noise environments, standard maximum likelihood objectives are often sufficient to capture the essential dynamics. Consequently, the marginal utility of value alignment naturally diminishes as the gap between reconstruction fidelity and decision-making utility is inherently narrower in DMC. To further evaluate the ability of value-alignment regularization to filter out interference from irrelevant tasks, we conducted a series of controlled experiments on Hopper Hop and Acrobot Swingup by inserting varying ratios of Gaussian noise into the proprioceptive input state dimensions.

*Table 3.* Robustness to noise in proprioceptive inputs tasks.

| Noise Ratio | DreamerV3 (2023 v1) | | DreamerV3+Var (Ours) | |
| --- | --- | --- | --- | --- |
| | Hopper Hop | Acrobot Swingup | Hopper Hop | Acrobot Swingup |
| Insert 0% noise | 236 | 261 | 238 | 295 |
| Insert 10% noise | 224(↓5.1%) | 238(↓8.8%) | 231(↓2.9%) | 284(↓3.7%) |
| Insert 20% noise | 197(↓16.5%) | 204(↓21.8%) | 219(↓7.9%) | 272(↓7.8%) |
| Insert 50% noise | 134(↓43.2%) | 124(↓52.5%) | 187(↓21.4%) | 241(↓18.3%) |

As shown in Tab.3, while the performance of both the standard DreamerV3 and our DreamerV3+Var degrades as the noise ratio increases, our approach demonstrates significantly higher resilience to these distractors. Specifically, under high-noise conditions (50% noise ratio), the standard DreamerV3 experiences a catastrophic performance collapse: its score plunges by 43.2% in Hopper Hop and 52.5% in Acrobot Swingup. In contrast, our DreamerV3+Var remains remarkably stable, with the performance drop restricted to 21.4% and 18.3%, respectively.

### 4.4. Visualization and Analysis of Imagined Trajectories

To qualitatively evaluate the modeling fidelity, we visualize the imagined trajectories generated by the world model. Fig.3 illustrates short-term comparisons in KungFuMaster and Qbert, while Fig.13 specifically depicts the long-term

evolution in Qbert. The rows represent the standard STORM (Zhang et al., 2023) (top), our STORM + Var (middle), and the ground truth (bottom). The results highlight the efficacy of value alignment in two critical dimensions:

**(1) Temporal Coherence and Ghosting Suppression:** In KungFuMaster Fig.3(a), the baseline STORM exhibits significant ghosting effects and motion blur. Conversely, our approach maintains sharp and distinct silhouettes of both the player and enemies. This enhanced dynamic integrity prevents the policy from being misled by hallucinations or blurred entities, thereby facilitating more robust planning.

**(2) Mitigating Object Vanishing in Long-Term Horizons:** In Qbert Fig.13, the standard STORM suffers from catastrophic object vanishing. In contrast, our STORM+Var effectively anchors the latent space to decision-critical entities. By value alignment, our model retains the character's presence and position over extended imagination horizons.

*Table 4.* Quantitative comparison of MSE and value error.

| KungFuMaster | MSE | Value error |
| --- | --- | --- |
| STORM (2023) | $3.2 \times 10^{-4}$ | 111.8 |
| STORM+Var (Ours) | $2.7 \times 10^{-4}$ | 38.9 |

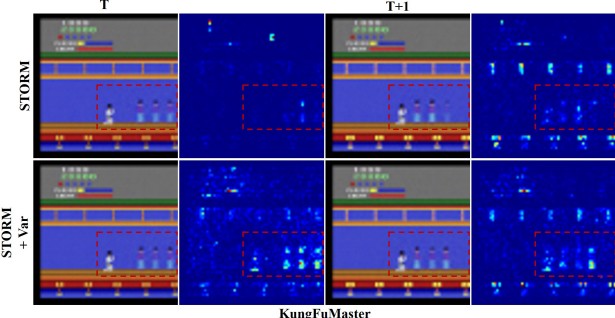

*Figure 4.* Feature saliency visualization on KungFuMaster game.

To further elucidate the underlying mechanism of value alignment, we conduct a joint analysis using feature saliency maps and quantitative metrics. Qualitatively, the saliency

analysis uncovers a significant redirection of the model's internal focus. As illustrated in Fig.4, activations in STORM + Var are highly concentrated on decision-critical targets, such as the agent and enemies, whereas the baseline's saliency remains sparse and fragmented. This visual evidence confirms that $L_{\text{var}}$ functions as a latent-space selective filter, enhancing feature precision by amplifying task-relevant signals while suppressing background redundancy. These observations are further substantiated by a quantitative evaluation across 3,120 sampled sequences (Tab.4): our method reduces the value error by approximately 65% (from 111.8 to 38.9) and simultaneously lowers the reconstruction MSE from $3.2 \times 10^{-4}$ to $2.7 \times 10^{-4}$. It demonstrates that value alignment enables a more efficient allocation of representational resources toward core entities, thereby enhancing both decision-making utility and dynamical integrity.

### 4.5. Generalizability Analysis of the Alignment Strategy

*Table 5.* Alignment strategy to reward and continuation signals.

| Game | DreamerV3 (2023 v1) | + Reward & Continue Alignment | STORM (2023) | + Reward & Continue Alignment |
|---|---|---|---|---|
| Alien | 876 | 1026(↑17.1%) | 1054 | 1152(↑9.3%) |
| BattleZone | 11138 | 11855(↑6.4%) | 7080 | 8062(↑13.9%) |
| Gopher | 1684 | 2248(↑33.5%) | 8551 | 12482(↑46.0%) |
| KungFuMaster | 16375 | 19388(↑18.4%) | 15760 | 18455(↑17.1%) |
| Qbert | 1224 | 1319(↑7.8%) | 2913 | 4146(↑42.3%) |

To assess the broad applicability of our framework, we extend latent-space alignment beyond the value function to reward and continuation signals. As shown in Tab.5, KL-based alignment between the prior and posterior distributions of these task-critical signals consistently improves both DreamerV3 and STORM. These results suggest that diverse objective-relevant signals can act as robust anchors for regularizing the latent space and guiding the world model toward decision-critical features. Overall, our approach is not merely a specific enhancement but a generalizable paradigm for synergizing maximum likelihood modeling with task-specific focus. It offers a lightweight, plug-and-play solution that can be seamlessly integrated into various MBRL methods to enhance their representational utility.

### 4.6. Ablation study

To evaluate the effectiveness of the adaptive weighting mechanism, we conduct an ablation study on the Atari 100k benchmark, comparing our method with a static-weight baseline ($\beta_{\text{var}} = 0.5$) under the same STORM backbone and warm-up setting. Ablation results in Tab.6 show that static weighting induces performance drops in tasks like Alien and DemonAttack. Conversely, the adaptive mechanism consistently achieves superior performance by dynamically balancing reconstruction fidelity and value alignment. Training curves (Fig.5) further reveal its self-regulating behavior: weights remain low during initial exploration to prioritize reconstruction, then increase as the model matures. This strat-

*Table 6.* Ablation study on the adaptive weighting.

| Atari Games | Alien | CrazyClimber | DemonAttack | BattleZone |
|---|---|---|---|---|
| STORM(2023) | 1054.3 | 47473.0 | 194.6 | 7080.0 |
| + static weight | 987.9(↓6.3%) | 55096.0(↑16.1%) | 180.3(↓7.3%) | 8480(↑19.8%) |
| + adaptive weight | 1361.4(↑29.1%) | 57335.0(↑20.8%) | 204.6(↑5.1%) | 10140(↑43.2%) |

*Figure 5.* Training curves of adaptive weighting mechanism.

egy reduces value-alignment loss without compromising representation convergence. Effectively, adaptive weighting acts as a learning curriculum that ensures dynamical stability before amplifying value guidance, thereby harmonizing training stability with decision-making utility.

We conduct tests on an NVIDIA 3090 GPU to evaluate the impact of value-alignment regularization on GPU memory and runtime. Tab.7 summarizes the effects on computational resources and training time. The results show that the computational overhead and training time introduced by value-alignment regularization are minimal, making their impact negligible relative to the overall algorithmic cost.

*Table 7.* Resource and computational overhead comparison.

| Methods | DreamerV3 (2023 v1) | DreamerV3+Var (Ours) | STORM (2023) | STORM+Var (Ours) |
|---|---|---|---|---|
| GPU Memory | 4560MB | 4626MB | 5806MB | 5864MB |
| total running time | 15.89h | 16.06h | 5.91h | 6.04h |
| Mean Score on Atari | 1.10 | 1.34 | 1.14 | 1.36 |

## 5. Conclusion

In this paper, we proposed a value-aligned world model that bridges the gap between maximum likelihood estimation and decision-making utility. By leveraging the structural stability of RSSM/TSSM backbones, our alignment regularization strategically redirects modeling capacity toward decision-critical features. To ensure stable optimization, a self-regulating adaptive weighting mechanism dynamically balances reconstruction fidelity with task-awareness. Extensive evaluations across 46 environments from the Atari 100k and DeepMind Control benchmarks demonstrate that our approach consistently enhances the performance of state-of-the-art methods. Notably, our algorithm achieves these gains as a lightweight, plug-and-play module with minimal computational overhead, offering a generalizable paradigm for improving representational ability in complex, high-dimensional environments.

## Acknowledgments

This work was supported by the National Natural Science Foundation of China under Grant 62325101.

## Impact Statement

This paper presents work whose goal is to advance the field of machine learning. There are many potential societal consequences of our work, none of which we feel must be specifically highlighted here.

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

# A. Agent behavior learning

Following DreamerV3(Hafner et al., 2023), both the critic and actor networks are trained using imagined trajectories generated by the world model. For environment interaction, actions are selected by sampling from the actor network without lookahead planning. In practice, both networks are implemented as MLPs, parameterized by $\theta$ and $\phi$, respectively.

$$\text{Critic Network:} \quad v_t \sim V_\theta(v_t|s_t) \qquad \text{Actor Network:} \quad a_t \sim \pi_\phi(a_t|s_t) \tag{6}$$

**Critic Learning:** In line with DreamerV3(Hafner et al., 2023), we estimate returns that incorporate rewards beyond the prediction horizon by computing bootstrapped $\lambda$-returns, which combine both predicted rewards and value estimates. The critic network is trained to predict the distribution of these $\lambda$-return estimates $R_t^\lambda$ by minimizing the maximum likelihood loss.

$$L_{critic} = \frac{1}{B \times T} \sum_{b=1}^{B} \sum_{t=1}^{T} -\ln p_\theta(R_T^\lambda|s_t) \quad R_t^\lambda = r_t + \gamma c_t\big((1-\lambda)v_t + \lambda R_{t+1}^\lambda\big) \tag{7}$$

**Actor Learning:** The actor network maximizes cumulative rewards using the REINFORCE(Williams, 1992) algorithm, with an added policy entropy loss to ensure sufficient exploration.

$$L_{actor} = \frac{1}{B \times T} \sum_{b=1}^{B} \sum_{t=1}^{T} -\text{sg}(A_t^\lambda)\log\pi_\phi(a_t|s_t) - \eta\text{H}\big[\pi_\phi(a_t|s_t)\big] \tag{8}$$

Here, $A_t^\lambda$ represents the advantage computed using normalized returns. To ensure stable learning, the returns are scaled using the exponentially moving average of the 5th and 95th percentiles of the batch.

$$A_t^\lambda = (R_t^\lambda - V_\theta(s_t))/\max(1, S) \quad S = \text{EMA}\big(\text{Per}(R_t^\lambda, 95) - \text{Per}(R_t^\lambda, 5), 0.99\big) \tag{9}$$

# B. Related work

Generally, most mainstream MBRL algorithms follow a two-stage training process: world model learning and behavior model learning. Depending on the strategy used to train the world model, MBRL algorithms can be further categorized into two types:

**Maximum likelihood world models** (Seo et al., 2023; Micheli et al., 2024) aim to accurately predict environmental dynamics from historical observations and actions, minimizing prediction errors via maximum likelihood estimation. PlaNet(Hafner et al., 2019b) introduces the Recurrent State-Space Model (RSSM), using recurrent neural networks (RNNs) and Variational Autoencoders (VAE)(Kingma & Welling, 2013) to model the world in latent space. Dreamer(Hafner et al., 2019a) builds on RSSM by incorporating an actor-critic framework that imagines behavior within the world model. DreamerV2(Hafner et al., 2020) optimizes this approach by replacing Gaussian latents with discrete categorical latents, improving stochastic dynamics representation. DreamerV3(Hafner et al., 2023) introduces structural and training modifications, enabling stable learning across various domains without hyperparameter tuning. Recent works have explored replacing RNN-based world models with Transformer architectures, incorporating self-attention mechanisms. TWM(Robine et al., 2023) proposes the Transformer State-Space Model (TSSM), treating states, actions, and rewards as independent tokens for dynamic modeling, while STORM(Zhang et al., 2023) integrates states and actions into a single token, enhancing training efficiency. More recently, DIAMOND(Alonso et al., 2024) introduced diffusion models for precise visual detail prediction, and TWISTER(Burchi & Timofte, 2025) applied Contrastive Predictive Coding in TSSM to model temporal dependencies. Despite these advancements, maximum likelihood world models still struggle with misalignment between the world model's training objectives and the policy optimization goal. Additionally, the need for each state precise prediction in maximum likelihood estimation limits the model's ability to effectively reconstruct task-relevant states, hindering its applicability in complex environments.

**Value-aware world models**, as the name suggests, aim to guide the world model with the value function to minimize the one-step value estimation error. The concept of Value-Aware Model Learning (VAML) was first introduced by (Farahmand et al., 2017), and IterVAML(Farahmand, 2018) was subsequently developed to iteratively optimize the policy and mitigate the "max-min" issue inherent in VAML. VaGraM(Voelcker et al., 2022) further enhances VAML by introducing Value-gradient weighted Model Learning, focusing the model on states that significantly influence the policy. More recently, CVAML(Voelcker et al.) introduces a variance correction term to address "overconfidence" in stochastic environments. While value-aware world models provide an intuitive approach to address the misalignment issue inherent in maximum likelihood world models, the instability of value estimation for out-of-distribution samples and the non-convexity of the VAML loss function make these algorithms susceptible to local optima during training. This, in turn, complicates their practical deployment and results in suboptimal performance compared to maximum likelihood-based methods. Moreover, these algorithms have not demonstrated strong empirical performance in complex, high-dimensional visual environments, such as Atari games.

Recent works have explored various strategies to integrate maximum likelihood and value-aware objectives. One prominent paradigm involves explicit joint optimization through auxiliary prediction heads, as seen in MuZero(Schrittwieser et al., 2020) and MuDreamer(Burchi & Timofte, 2024). MuZero learns world model dynamics solely through value, reward, and policy targets, discarding pixel-level reconstruction entirely to focus on planning-relevant transitions. While effective, it lacks the generative stability and visual grounding provided by reconstruction. MuDreamer attempts to bridge this by incorporating an auxiliary value branch into reconstruction-based RSSM frameworks. However, as noted in Fig.1(a), these explicit multi-head structures often suffer from gradient dominance, where high-dimensional reconstruction gradients overshadow sparse value signals, necessitating delicate hyperparameter tuning or complex scheduling to maintain a balance. Another approach utilizes meta-learning to bridge the two paradigms. For example, TEMPO (Yuan et al., 2023) introduces a bi-level framework, adding a meta-weighting network atop the maximum-likelihood model to generate sample weights that minimize task-aware model loss. While TEMPO shows promising results, its bi-level structure significantly increases computational complexity, inference time, and resource consumption, making practical deployment challenging. More recently, inspired by the rise of pre-trained large models, some approaches leverage external prior knowledge for decision-sensitive reconstruction. PSP (Hutson et al., 2024) incorporates a pre-trained segmentation model, enabling the world model to capture key environmental features. Zhang et al. (2026) assigns higher optimization weights to decision-relevant regions using object detection, while DreamVLA (Zhang et al., 2025b) integrates 3D knowledge and semantic segmentation to improve world model predictions. Despite their performance gains, the high computational demands of large models limit training efficiency. Furthermore, the potential misalignment between fixed pre-trained priors and specific downstream tasks can complicate effective model optimization.

In contrast to these approaches, our proposed Latent-Space Value Alignment offers a minimalist and intrinsic solution. By imposing value constraints directly within the latent distribution transitions, we circumvent the need for complex auxiliary branches, meta-architectures, or heavy external models, achieving superior synergy with negligible overhead.

# C. More experiments

## C.1. Extended Analysis on Noise Robustness and Quantitative Evidence

*Table 8.* Noise-robustness results on Walker Run with varying ratios of Gaussian noise injected into proprioceptive input dimensions.

| Noise Ratio | DreamerV3 | DreamerV3+Var |
|---|---|---|
| 0% | **726** | 716 |
| 10% | 675 | **694** |
| 20% | 593 | **657** |
| 50% | 426 | **569** |

To provide broader empirical evidence for Value-Alignment Regularization (Var), we extend our analysis from two complementary perspectives: robustness to task-irrelevant noise and quantitative representation quality. For noise robustness, we evaluate DreamerV3 and DreamerV3+Var on Walker Run by injecting varying ratios of Gaussian noise into proprioceptive input dimensions. As shown in Table 8, DreamerV3+Var consistently achieves higher scores under noisy observations, and the advantage becomes more pronounced as the noise ratio increases. In particular, under 50% inserted noise, DreamerV3+Var improves the score from 426 to 569, suggesting that value alignment helps suppress task-irrelevant perturbations and preserve decision-critical information.

*Table 9.* Quantitative evaluation of reconstruction quality and value prediction accuracy across Atari environments.

| Environment | Method | MSE $\downarrow$ | Value Error $\downarrow$ |
|---|---|---|---|
| CrazyClimber | STORM | 0.00126 | 233.48 |
| | STORM+Var | **0.00119** | **64.67** |
| Krull | STORM | 0.00073 | 110.96 |
| | STORM+Var | **0.00059** | **16.44** |
| Gopher | STORM | 0.00040 | 32.40 |
| | STORM+Var | **0.00033** | **6.68** |
| UpNDown | STORM | 0.00447 | 80.69 |
| | STORM+Var | **0.00396** | **6.53** |

We further report quantitative evidence on CrazyClimber, Krull, Gopher, and UpNDown in Table 9. Across all environments, STORM+Var consistently reduces both reconstruction MSE and value prediction error compared with STORM. The improvement is especially substantial in value error, while reconstruction fidelity is also preserved or improved. These results indicate that Var enhances the value relevance of latent representations without sacrificing reconstruction quality, further supporting its effectiveness in guiding world models toward task-critical features.

## C.2. Ablation on Static and Adaptive Alignment Weights

*Table 10.* Ablation study on static and adaptive alignment weights.

| Method | Alien | CrazyClimber | DemonAttack | BattleZone |
|---|---|---|---|---|
| STORM | 1054 | 47473 | 195 | 7080 |
| + static $\beta_{var} = 0.1$ | 1124 | 49465 | 198 | 7860 |
| + static $\beta_{var} = 0.3$ | 1253 | 51785 | 191 | 7590 |
| + static $\beta_{var} = 0.5$ | 988 | 55096 | 180 | 8480 |
| + static $\beta_{var} = 1.0$ | 923 | 53260 | 167 | 9165 |
| + adaptive $\beta_{var}$ | **1361** | **57335** | **205** | **10140** |

We further evaluate whether the improvement depends on a specific alignment weight by comparing our adaptive scheme with four fixed coefficients, $\beta_{var} \in \{0.1, 0.3, 0.5, 1.0\}$. As shown in Table 10, static weights can improve over STORM in some cases, but their effects are task-dependent. Larger weights may benefit BattleZone but degrade Alien and DemonAttack, indicating that excessive alignment pressure can interfere with dynamics representation learning. In contrast, adaptive weighting achieves the best performance across all evaluated tasks, suggesting that dynamically adjusting the alignment strength provides a more robust balance between value alignment and dynamics learning.

## C.3. Complementarity with Explicit Replay Value Prediction

We further investigate the relationship between latent-space Value-Alignment Regularization (Var) and explicit replay value prediction. Although replay value loss can encourage the posterior representation to encode task-relevant information, it suffers from two inherent limitations. First, the posterior is jointly optimized by multiple prediction heads, including reconstruction, reward, and continuation heads. As a result, the scalar value-prediction signal can be dominated by high-dimensional reconstruction gradients, making value-aware representation learning less stable. Second, even when the posterior captures value-relevant features, the standard dynamics loss does not explicitly prioritize their transfer to the prior, since it treats all latent dimensions indiscriminately.

In contrast, Var operates directly in the latent space by aligning the value-aware structure between the posterior and the prior. This design avoids direct gradient competition among different prediction heads and explicitly encourages the prior to preserve decision-critical information encoded in the posterior. Therefore, replay value loss and Var introduce value awareness from different perspectives: the former shapes the posterior through explicit value prediction, while the latter regularizes the prior-posterior relationship through latent-space alignment. These two mechanisms are thus orthogonal and potentially complementary.

To empirically examine this complementarity, we conduct experiments under four configurations based on the DreamerV3-v2 framework: (a) the baseline without value-aware components, (b) the baseline with replay value loss, (c) the baseline with Var, and (d) the baseline with both replay value loss and Var. The results are reported in Table 11.

*Table 11.* Comparison between explicit replay value loss and latent-space Value-Alignment Regularization (Var) on Atari tasks. We evaluate four configurations: (a) baseline, (b) baseline with replay value loss, (c) baseline with Var, and (d) baseline with both replay value loss and Var.

| Method | Alien | Gopher | Krull | KungFuMaster |
|---|---|---|---|---|
| (a) Baseline | 1049 | 2393 | 8040 | 25842 |
| (b) + Replay value loss | 942 | 2167 | 9243 | 26220 |
| (c) + Var | **1406** | 2726 | 9769 | 26564 |
| (d) + Replay value loss + Var | 1194 | **2947** | **11303** | **27586** |

As shown in Table 11, replay value loss alone leads to inconsistent improvements and can even degrade performance on some tasks, such as Alien and Gopher. In comparison, Var provides more stable gains across all evaluated environments, suggesting that latent-space alignment is a more robust mechanism for introducing value awareness into world-model learning. Moreover, combining replay value loss with Var achieves the best performance on three out of four tasks, including Gopher, Krull, and KungFuMaster. These results indicate that explicit value prediction and latent-space value alignment are not mutually exclusive. Instead, they can provide complementary value-aware signals, with Var serving as a more stable regularization mechanism and replay value loss offering additional benefits in several tasks.

## C.4. Robustness under Limited Model Capacity after 1M-Step Convergence

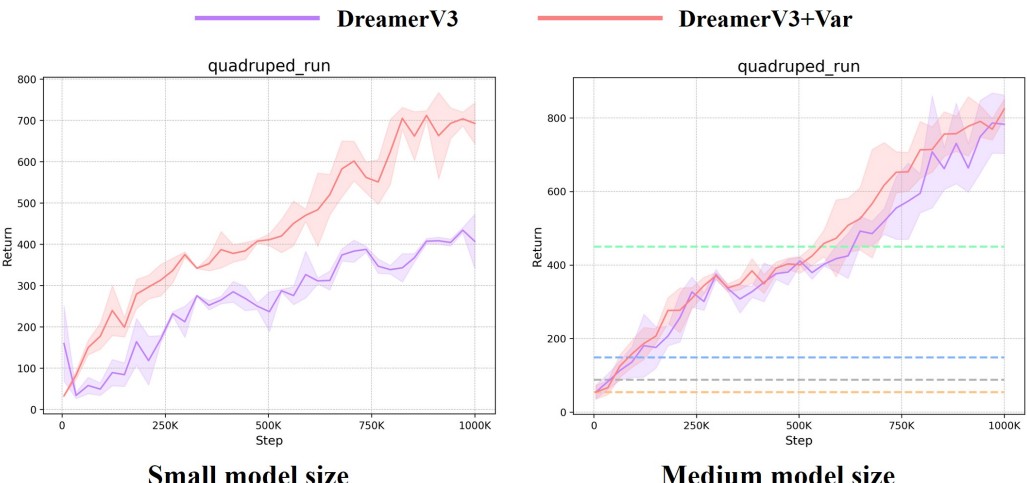

*Figure 6.* Results on DMC Quadruped run with different model sizes.

To further evaluate the efficacy of Value-Alignment Regularization (Var) under varying model capacities, we conducted experiments on the Quadruped Run environment within the DeepMind Control Suite. The training duration was extended to 1 million steps to ensure full convergence.

Figure 6 presents the training curves for both "Small" and "Medium" model configurations. In the Medium setting, where modeling capacity is relatively sufficient, the performance of our approach is on par with the vanilla DreamerV3. However, a significant performance gap emerges as representational resources are constrained. In the Small configuration, the vanilla DreamerV3 suffers from a notable degradation in convergence and final scores. Crucially, the incorporation of value alignment effectively compensates for this capacity deficit, enabling the model to maintain high decision-making effectiveness despite the reduced parameter count. These results demonstrate that value alignment promotes a more efficient allocation of limited representational resources toward task-critical features, thereby enhancing the model's resilience in resource-constrained scenarios.

## D. Generalizing Latent-Space Alignment to Reward and Continuation Signals

To examine whether latent-space alignment is specific to value prediction or can be generalized to other task-relevant signals, we instantiate the alignment objective with reward and continuation predictions as alternative anchors. In this variant, the value-alignment loss is replaced by reward and continuation alignment, rather than being used jointly with them. This design allows us to isolate whether behaviorally grounded signals beyond value estimates can also regularize the prior-posterior relationship and improve the task relevance of learned latent representations.

For reward alignment, the reward predictor outputs a discrete distribution. We therefore minimize the KL divergence between the reward distributions predicted from the posterior and prior latent states:

$$\mathcal{L}_{\text{rar}} = D_{\text{KL}} \left[ p_\alpha(r_t \mid s_t) \parallel p_\alpha(\tilde{r}_t \mid \tilde{s}_t) \right], \tag{10}$$

where $s_t$ and $\tilde{s}_t$ denote the posterior and prior latent states, respectively. For continuation alignment, since the continuation predictor outputs a scalar probability, we use mean squared error to align the continuation predictions:

$$\mathcal{L}_{\text{car}} = \text{MSE} \left[ p_\alpha(c_t \mid s_t), p_\alpha(\tilde{c}_t \mid \tilde{s}_t) \right]. \tag{11}$$

The resulting reward-continuation alignment objective is defined as

$$\mathcal{L}_{\text{rc-align}} = \mathcal{L}_{\text{rar}} + \mathcal{L}_{\text{car}}. \tag{12}$$

The world-model objective for this variant is then

$$\mathcal{L}_{\text{wm}} = \mathcal{L}_{\text{base}} + \beta_{\text{align}} \mathcal{L}_{\text{rc-align}}, \tag{13}$$

where $\mathcal{L}_{\text{base}}$ denotes the standard world-model learning objective. We use the same warm-up strategy and adaptive weighting schedule as value alignment, with $\beta_{\text{align}}$ following the original $\beta_{\text{var}}$ schedule.

This formulation shows that the proposed alignment principle is not tied to the value function alone. Reward and continuation signals can also serve as effective anchors for injecting task-relevant structure into the latent space, suggesting that latent-space alignment provides a general mechanism for improving the representational utility of world models.

# E. More discussion on adaptive weight $\beta_{var}$

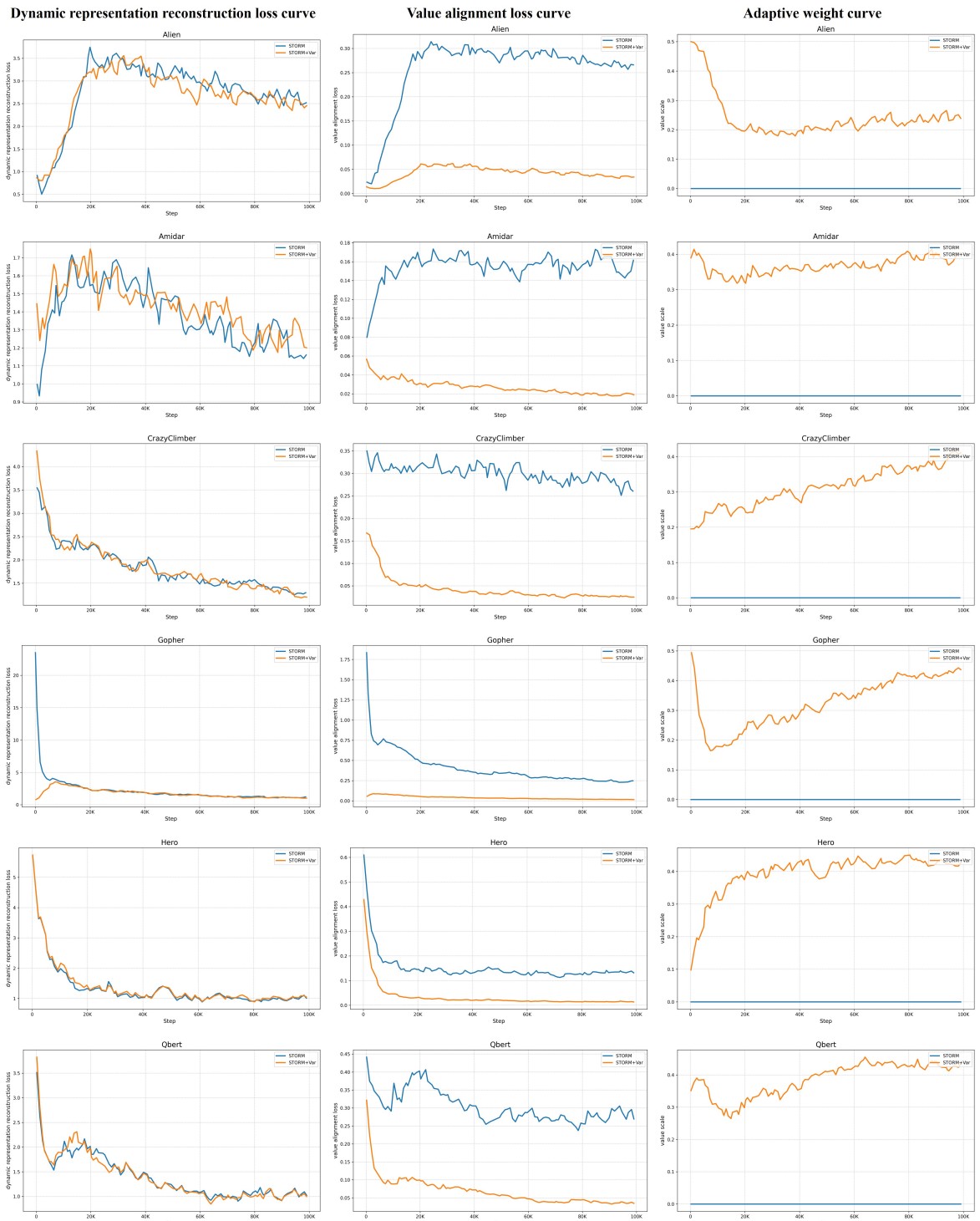

*Figure 7.* Dynamic reconstruction loss, value alignment loss and adaptive weight curve.

To visually substantiate the "self-regulating" nature of our framework, we visualize the training trajectories of the reconstruction loss ($L_{dyn}$), the value-alignment loss ($L_{var}$), and the adaptive weight ($\beta_{var}$) across six diverse Atari games (Fig. 7).

**Dynamic Modulation of Alignment Intensity.** As illustrated in the third column of Fig.7, the adaptive weight $\beta_{\text{var}}$ undergoes complex dynamic adjustment. In environments like Gopher, Hero, and Qbert, $\beta_{\text{var}}$ exhibits a clear upward trajectory as the world model begins to master environmental dynamics (evidenced by the declining $L_{\text{dyn}}$). In more challenging scenarios like Alien, the weight curve shows a more nuanced modulation, stabilizing after an initial adjustment phase. This confirm that our adaptive mechanism effectively functions as a dynamic curriculum: it throttles value guidance when environmental modeling is volatile and intensifies it as the dynamical foundation becomes secure.

**Consistent Suppression of Value-Alignment Error.** The second column reveals a striking disparity between the baseline and our method. Across all games, STORM+Var (orange) maintains a significantly lower and more stable value-alignment loss compared to the vanilla STORM (blue). This indicates that our regularization term $L_{\text{var}}$ effectively compels the latent state to anchor on value-sensitive features, bridging the gap between representation learning and decision-making utility throughout the entire training process.

**Synergy Without Interference.** Crucially, the first column shows that the $L_{\text{dyn}}$ curves for STORM+Var remain nearly identical to, or in some cases slightly lower than, those of the baseline. This observation provides definitive proof that maximum likelihood optimization and value alignment are not antagonistic. Instead, they achieve a synergistic harmony: by filtering out task-irrelevant background noise through value guidance, the world model can actually allocate its representational capacity more efficiently, preserving, or even enhancing, the integrity of the environmental dynamics.

## F. Discussion on potential value-delusional states

**Dynamic representation reconstruction loss curve**          **Value alignment loss curve**

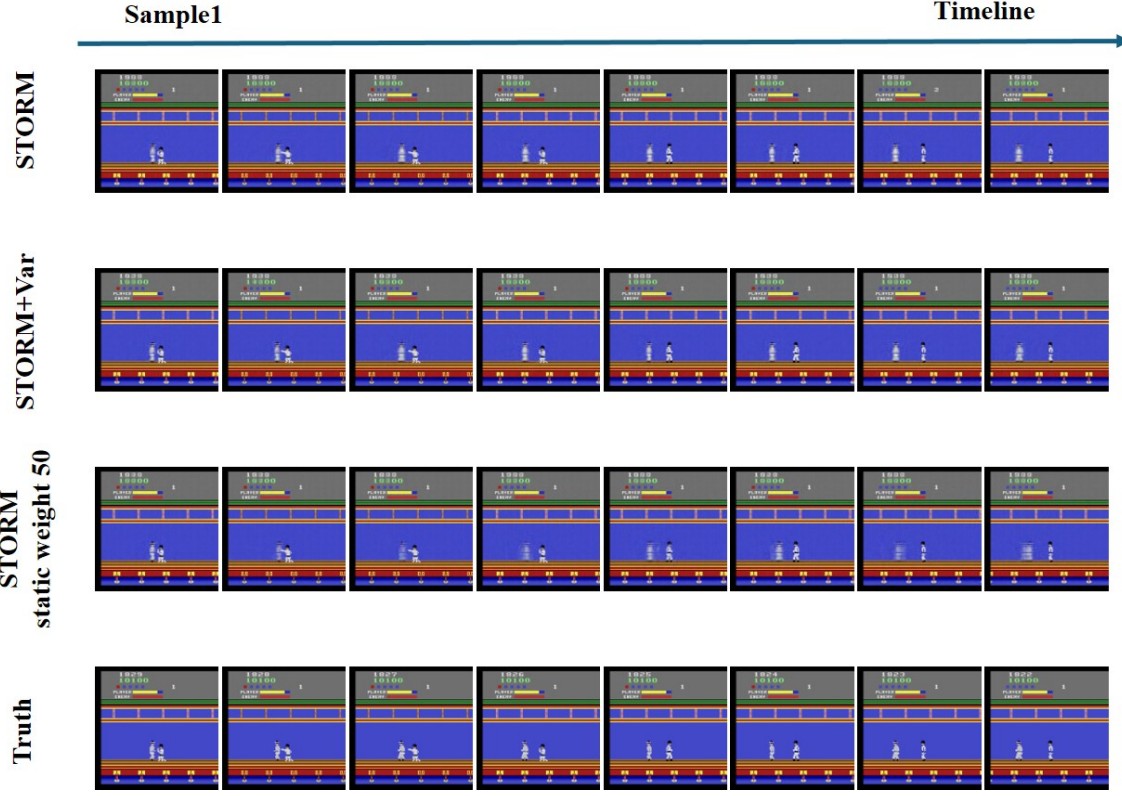

*Figure 8.* Dynamic reconstruction loss and value alignment loss curve during training.

To investigate the potential pitfalls of over-prioritizing value awareness, we examine the risk of the world model generating "hallucinated" or dynamically inconsistent states. This occurs when the optimization excessively favors the value-alignment term at the expense of dynamical integrity.

Fig.8 illustrates the training trajectories of the dynamics loss ($L_{\mathrm{dyn}}$) and value-alignment loss ($L_{\mathrm{var}}$) in KungFuMaster. We compare the original STORM (blue), STORM+Var with adaptive weighting (orange), and an aggressively aligned version with a large static weight ($\beta_{\mathrm{var}} = 50$, green). The curves reveal that while the aggressive static weight achieves the lowest $L_{\mathrm{var}}$, it triggers a significant degradation in $L_{\mathrm{dyn}}$. This suggests that the model is sacrificing physical consistency to satisfy value constraints.

*Figure 9.* World model visualization result comparisons.

The visual results in Fig.9 and Fig.10 substantiate this trade-off. Under excessive value alignment (green setting), the

world model produces pronounced reconstruction artifacts and hallucinations, such as unstable "ghosting" of enemies or the sudden manifestation of entities out of thin air. These delusional states, while easy for the value function to interpret, violate the underlying environmental dynamics. Crucially, our adaptive weighting mechanism effectively navigates this delicate trade-off, dynamically regulating the balance to ensure that value guidance enhances representational utility without compromising the foundational dynamical integrity of the RSSM.

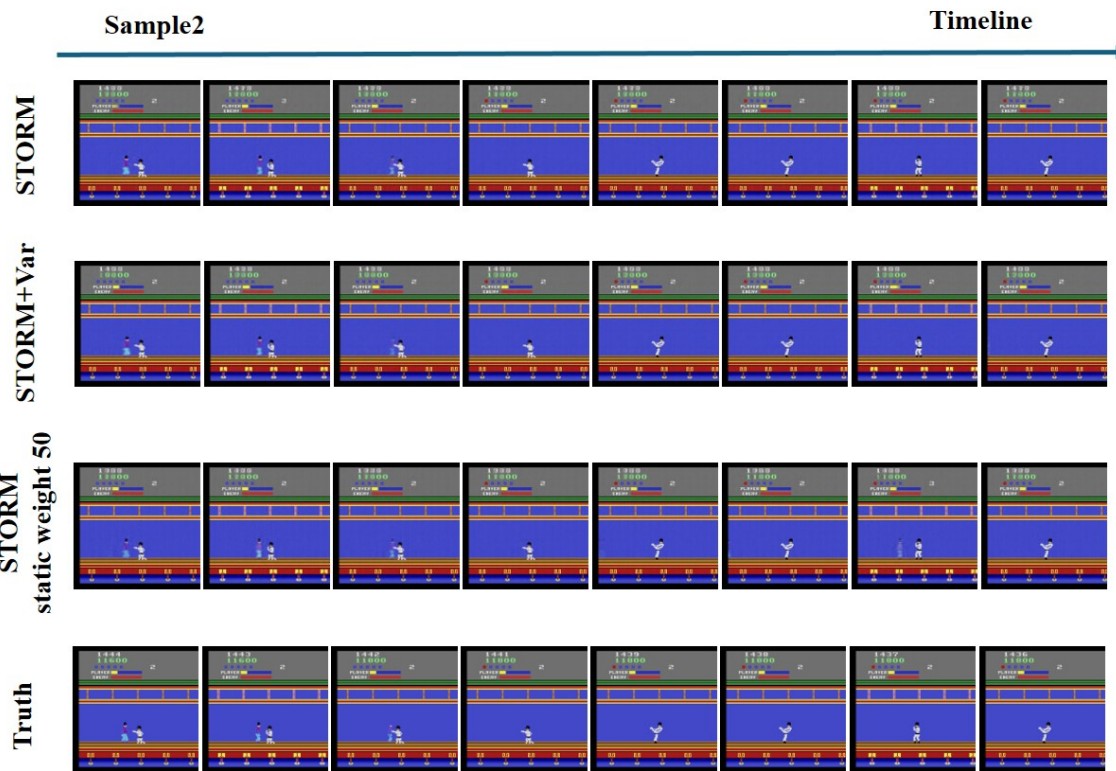

*Figure 10.* World model visualization result comparisons.

# G. Training curves across various benchmarks

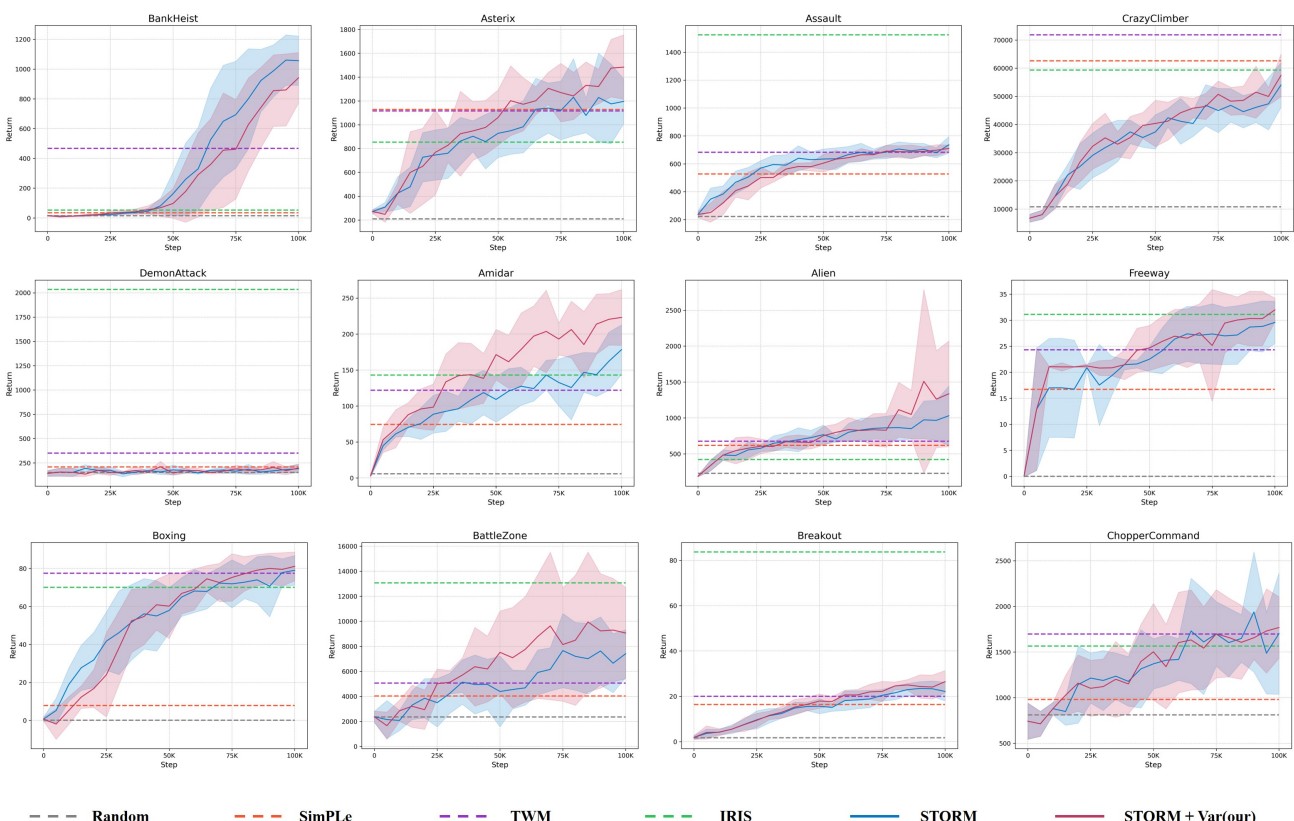

*Figure 11.* Training curve on Atari 100k(Kaiser et al., 2019; Robine et al., 2023; Micheli et al., 2022; Zhang et al., 2023).

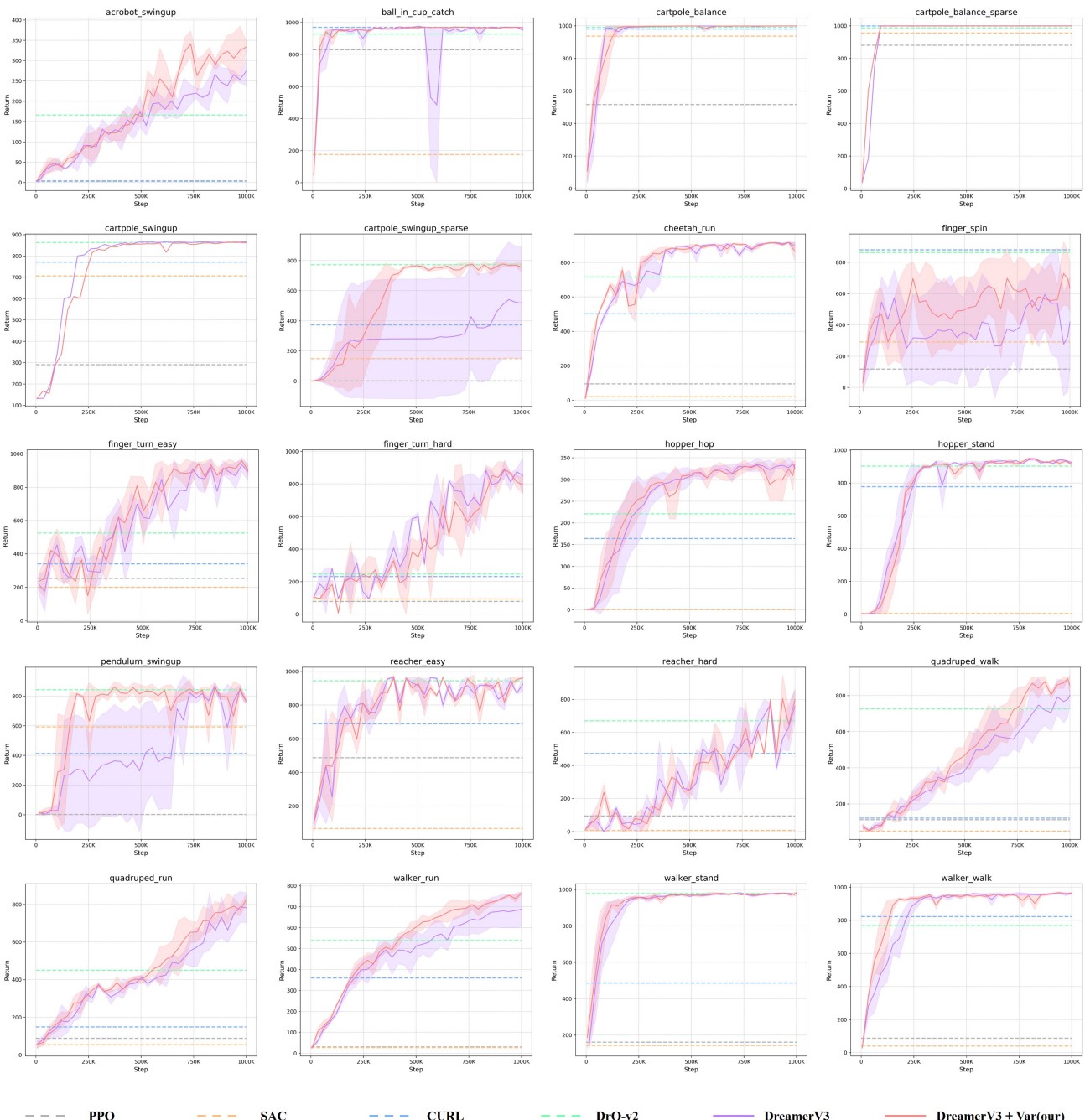

*Figure 12.* Training curve on DMC suite(Schulman et al., 2017; Haarnoja et al., 2018; Laskin et al., 2020; Yarats et al., 2021; Hafner et al., 2023).

# H. Long-term imagined trajectories

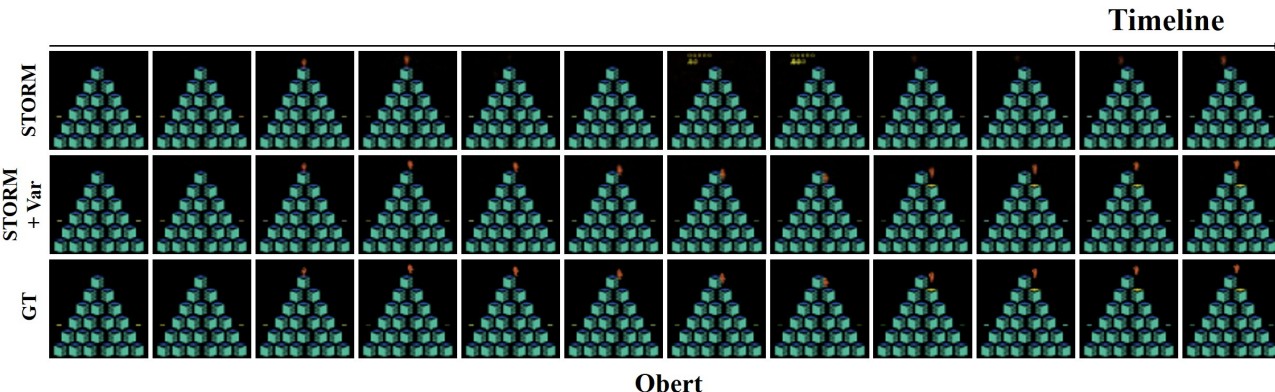

*Figure 13.* Long-term imagined trajectories from the world model in Qbert games.

## I. Original reported and reproduced results on the Atari 100k.

*Table 12.* DreamerV3 original and reproduced quantitative results on the Atari 100k benchmark.

| Game | Random | Human | DreamerV3 v1 reported | DreamerV3 v2 reported | DreamerV3 our reproduced | DreamerV3+Var our reproduced |
|---|---|---|---|---|---|---|
| Alien | 227.8 | 7127.7 | 959 | 1118 | 875.88 | 1233.2 |
| Amidar | 5.8 | 1719.5 | 139 | 97 | 143.7 | 185.4 |
| Assault | 222.4 | 742.0 | 706 | 683 | 843.7 | 981.38 |
| Asterix | 210.0 | 8503.3 | 932 | 1062 | 1102.5 | 1162.6 |
| BankHeist | 14.2 | 753.1 | 649 | 398 | 1072.0 | 1121.2 |
| BattleZone | 2360.0 | 37187.7 | 12250 | 20300 | 11138.0 | 12750.0 |
| Boxing | 0.1 | 12.1 | 78 | 82 | 80.3 | 87.4 |
| Breakout | 1.7 | 30.5 | 31 | 10 | 25.3 | 45.6 |
| ChopperCommand | 811.0 | 7387.8 | 420 | 2222 | 1438.0 | 1826.0 |
| CrazyClimber | 10780.5 | 35829.4 | 97190 | 86225 | 89900.0 | 81720.0 |
| DemonAttack | 152.1 | 1971.0 | 303 | 577 | 223.9 | 227.2 |
| Freeway | 0.0 | 29.6 | 0 | 0 | 30.2 | 31.6 |
| Frostbite | 65.2 | 4334.7 | 909 | 3377 | 1628.0 | 347.9 |
| Gopher | 257.6 | 2412.5 | 3730 | 2160 | 1683.9 | 2807.0 |
| Hero | 1027.0 | 30826.4 | 11161 | 13354 | 4994.4 | 9360.6 |
| Jamesbond | 29.0 | 302.8 | 445 | 540 | 332.0 | 542.0 |
| Kangaroo | 52.0 | 3035.0 | 4098 | 2643 | 1529.2 | 3650.4 |
| Krull | 1598.0 | 2665.5 | 7782 | 8171 | 8364.8 | 9821.4 |
| KungFuMaster | 258.5 | 22736.3 | 21420 | 25900 | 16375.0 | 21075.0 |
| MsPacman | 307.3 | 6951.6 | 1327 | 1521 | 1947.0 | 1749.5 |
| Pong | -20.7 | 14.6 | 18 | -4 | 19.1 | 19.8 |
| PrivateEye | 24.9 | 69571.3 | 882 | 3238 | 2331.2 | -115.6 |
| Qbert | 163.9 | 13455 | 3405 | 2921 | 1223.5 | 2267.8 |
| RoadRunner | 11.5 | 7845.0 | 15565 | 19230 | 9868.6 | 14704.0 |
| Seaquest | 68.4 | 42054.7 | 618 | 962 | 513.2 | 546.3 |
| UpNDown | 533.4 | 11693.2 | 7567 | 46910 | 12679.2 | 18485.4 |
| Mean (↑) | 0.00 | 1.00 | 1.12 | 1.25 | 1.10 | 1.34 |
| Median (↑) | 0.00 | 1.00 | 0.49 | 0.49 | 0.58 | 1.00 |

Table 12 presents the original recorded results for both the v1 and v2 versions of DreamerV3, alongside our reproduced results using the PyTorch implementation, on the Atari 100K dataset.

*Table 13.* STORM original and reproduced quantitative results on the Atari 100k benchmark.

| Game | Random | Human | STORM ori reported | STORM reproduced by (Meo et al., 2024) | STORM our reproduced | STORM+Var our reproduced |
|---|---|---|---|---|---|---|
| Alien | 227.8 | 7127.7 | 984 | 1364 | 1054.3 | 1361.4 |
| Amidar | 5.8 | 1719.5 | 205 | 239 | 177.29 | 248.36 |
| Assault | 222.4 | 742.0 | 801 | 707 | 715.9 | 752.55 |
| Asterix | 210.0 | 8503.3 | 1028 | 865 | 1276.0 | 1535.0 |
| BankHeist | 14.2 | 753.1 | 641 | 375 | 1060.5 | 935.0 |
| BattleZone | 2360.0 | 37187.7 | 13540 | 10780 | 7080.0 | 10140.0 |
| Boxing | 0.1 | 12.1 | 80 | 80 | 78.6 | 83.0 |
| Breakout | 1.7 | 30.5 | 16 | 12 | 20.88 | 26.43 |
| ChopperCommand | 811.0 | 7387.8 | 1888 | 2293 | 1768.0 | 1695.0 |
| CrazyClimber | 10780.5 | 35829.4 | 66776 | 54707 | 47473.0 | 57335.0 |
| DemonAttack | 152.1 | 1971.0 | 165 | 229 | 194.6 | 204.6 |
| Freeway | 0.0 | 29.6 | 33.5 | 0 | 29.7 | 32.0 |
| Frostbite | 65.2 | 4334.7 | 1316 | 646 | 258.8 | 260.2 |
| Gopher | 257.6 | 2412.5 | 8240 | 2631 | 8551.0 | 13509.6 |
| Hero | 1027.0 | 30826.4 | 11044 | 11044 | 12249.2 | 12574.0 |
| Jamesbond | 29.0 | 302.8 | 509 | 552 | 446.4 | 462.5 |
| Kangaroo | 52.0 | 3035.0 | 4208 | 1716 | 1542.0 | 3322.6 |
| Krull | 1598.0 | 2665.5 | 8412 | 6869 | 8360.1 | 8896.5 |
| KungFuMaster | 258.5 | 22736.3 | 26182 | 20144 | 15760 | 26615.0 |
| MsPacman | 307.3 | 6951.6 | 2674 | 2673 | 1906.9 | 2417.3 |
| Pong | -20.7 | 14.6 | 11 | 8 | 20.6 | 20.2 |
| PrivateEye | 24.9 | 69571.3 | 7781 | 2734 | 414.4 | 2584.7 |
| Qbert | 163.9 | 13455 | 4523 | 2986 | 2912.5 | 4243.4 |
| RoadRunner | 11.5 | 7845.0 | 17564 | 12477 | 11523.0 | 13999.0 |
| Seaquest | 68.4 | 42054.7 | 525 | 525 | 441.4 | 430.0 |
| UpNDown | 533.4 | 11693.2 | 7985 | 7985 | 6406.4 | 8982.6 |
| Mean (↑) | 0.00 | 1.00 | 1.27 | 0.95 | 1.14 | 1.36 |
| Median (↑) | 0.00 | 1.00 | 0.58 | 0.36 | 0.51 | 0.81 |

Table 13 presents the original recorded results for STORM, third-party(Meo et al., 2024) reproduction results, and our reproduced results using the official STORM codebase on the Atari 100K dataset.

## J. Code and Declarations

We implement our method on the Torch-version DreamerV3 code (https://github.com/NM512/dreamerv3-torch) and official STORM code (https://github.com/weipu-zhang/STORM). For detailed code implementation, please refer to the supplementary materials. In the Freeway environment of Atari 100k, we applied the same trick as used in IRIS(Micheli et al., 2022; 2024).

In our work, the large language model is used solely for text refinement and grammar correction, with no other applications.

## K. Detailed model structure and hyperparameter

### K.1. STORM

The network architecture and parameters for the STORM model are consistent with those in (Zhang et al., 2023). The specific architecture and parameters are detailed below.

*Table 14.* Image Encoder Architecture and Parameters: The image encoder takes an input image of size $3 \times 64 \times 64$ and consists of four convolutional blocks, followed by Flatten, Linear, and Reshape layers. Each convolutional block is composed of a Conv layer, a BN layer, and a ReLU activation function. The Conv layer (LeCun et al., 1989) has a kernel size of 4, a stride of 2, and a padding of 1. The BN layer (Ioffe & Szegedy, 2015) is used for batch normalization. The Flatten and Reshape layers are used to adjust the tensor indexing.

| Module | Output Tensor Shape |
|---|---|
| Input: Environment Image ($o_t$) | $3 \times 64 \times 64$ |
| Convolutional Block 1 (Conv + BN + ReLU) | $32 \times 32 \times 32$ |
| Convolutional Block 2 (Conv + BN + ReLU) | $64 \times 16 \times 16$ |
| Convolutional Block 3 (Conv + BN + ReLU) | $128 \times 8 \times 8$ |
| Convolutional Block 4 (Conv + BN + ReLU) | $256 \times 4 \times 4$ |
| Flatten | 4096 |
| Linear | 1024 |
| Reshape | $32 \times 32$ |
| Output: distribution ($\mathcal{Z}_t$) | $32 \times 32$ |

*Table 15.* Image Decoder Architecture and Parameters: The image decoder takes a $32 \times 32$ sampled value, $z_t$, as input. The network architecture consists of DeConv modules, which are composed of a DeConv layer (Zeiler et al., 2010), a BN layer, and a ReLU activation function. The DeConv layers have a kernel size of 4, a stride of 2, and a padding of 1.

| Module | Output Tensor Shape |
|---|---|
| Input: Random Sample ($z_t$) | $32 \times 32$ |
| Flatten | 1024 |
| Linear + BN + ReLU | 4096 |
| Reshape | $256 \times 4 \times 4$ |
| DeConv Block 1 (DeConv + BN + ReLU) | $128 \times 8 \times 8$ |
| DeConv Block 2 (DeConv + BN + ReLU) | $64 \times 16 \times 16$ |
| DeConv Block 3 (DeConv + BN + ReLU) | $32 \times 32 \times 32$ |
| DeConv | $3 \times 64 \times 64$ |
| Output: Decoded Image ($\hat{o}_t$) | $3 \times 64 \times 64$ |

*Table 16.* Action Mixer Architecture and Parameters: The Action mixer takes a $32 \times 32$ sampled value, $z_t$, and a A-dimensional action as input (where the action dimension varies from 3 to 18 depending on the game). Concatenate merges the last dimension of the two tensors. D is the feature dimension of the Transformer. LN denotes layer normalization (Ba et al., 2016).

| Module | Output Tensor Shape |
|---|---|
| Input: Random Sample ($z_t$), Action ($a_t$) | $32 \times 32$, A |
| Reshape and concatenate | 1024 + A |
| Linear + LN + ReLU | D |
| Linear2 + LN2 | D |
| Output: $e_t$ | D |

*Table 17.* Positional Encoding Module: The Positional Encoding Module adds a learnable parameter matrix, $w_{1:T}$, to the input tensor, $e_{1:T}$. The operation is represented as $e_{1:T} + w_{1:T}$, where the sequence length is denoted by $T$ and the feature dimension by $D$. The matrix $w_{1:T}$ has a shape of $T \times D$. Following the addition, Layer Normalization (LN) is applied.

| Module | Output Tensor Shape |
|---|---|
| Input: $e_{1:T}$ | $T \times D$ |
| Add + LN | $T \times D$ |
| Output: $x$ | $T \times D$ |

*Table 18.* Transformer Module

| Module | Sub-Module | Output Tensor Shape |
|---|---|---|
| Input | $x$ | $T \times D$ |
| MHSA | Multi-head self attention | $T \times D$ |
|  | Linear + Dropout | $T \times D$ |
|  | Residual | $T \times D$ |
|  | LN | $T \times D$ |
| FFN | Linear + ReLU | $T \times 2D$ |
|  | Linear + Dropout | $T \times D$ |
|  | Residual | $T \times D$ |
|  | LN | $T \times D$ |
| Output: | $h_{1:T}$ | $T \times D$ |

*Table 19.* Transformer-Based Sequence Model Architecture and Parameters: The Positional encoding module is defined in the Table 17 for Positional encoding module. The Transformer block module is defined in the Table 18 for Transformer module.

| Module | Output Tensor Shape |
|---|---|
| Input: $e_{1:T}$ | $T \times D$ |
| Positional encoding | $T \times D$ |
| Transformer blocks $\times$K | $T \times D$ |
| Output: $h_{1:T}$ | $T \times D$ |

*Table 20.* Other MLP Modules: This table details the architecture of other pure MLP modules.

| Module | Number of MLP Layers | Input Dim | Hidden Dim | Output Dim |
|---|---|---|---|---|
| Dynamics head $g_\phi^D$ | 1 | D | / | 1024 |
| Reward predictor $g_\phi^R$ | 3 | D | D | 255 |
| Continuation predictor $g_\phi^C$ | 3 | D | D | 1 |
| Policy network $\pi_\theta(a_t|s_t)$ | 3 | D | D | A |
| Critic network $V\psi(s_t)$ | 3 | D | D | 255 |

*Table 21.* STORM Network Hyperparameters.

| Hyperparameter | Symbol | Value |
|---|---|---|
| Transformer layers | $K$ | 2 |
| Transformer feature dimension | $D$ | 512 |
| Transformer heads | – | 8 |
| Dropout probability | $P$ | 0.1 |
| World model training batch size | $B_1$ | 16 |
| World model training batch length | $T$ | 64 |
| Imagination batch size | $B_2$ | 1024 |
| Imagination context length | $C$ | 8 |
| Imagination horizon | $L$ | 16 |
| Update world model every env step | – | 1 |
| Update agent every env step | – | 1 |
| Environment context length | – | 16 |
| Gamma | $\gamma$ | 0.985 |
| Lambda | $\lambda$ | 0.95 |
| Entropy coefficient | $\eta$ | $3 \times 10^{-4}$ |
| Critic EMA decay | $\sigma$ | 0.98 |
| Optimizer | – | Adam |
| World model learning rate | – | $1.0 \times 10^{-4}$ |
| World model gradient clipping | – | 1000 |
| Actor-critic learning rate | – | $3.0 \times 10^{-5}$ |
| Actor-critic gradient clipping | – | 100 |
| Gray scale input | – | False |
| Frame stacking | – | False |
| Frame skipping | – | 4 |
| Use of life information | – | True |

## K.2. DreamerV3

The DreamerV3 network architecture and parameters remain consistent with those detailed in (Hafner et al., 2023). Table 22 are the hyperparameters.

*Table 22.* DreamerV3 Network Hyperparameters

| Hyperparameter | Value |
| --- | --- |
| Replay capacity | $5 \times 10^6$ |
| Batch size | 16 |
| Batch length | 64 |
| Activation | RMSNorm+SiLU |
| Learning rate | $4 \times 10^{-5}$ |
| Gradient clipping | AGC(0.3) |
| Optimizer | LaProp($\epsilon = 10^{-20}$) |
| World Model reconstruction loss scale | 1 |
| World Model dynamics loss scale | 1 |
| World Model representation loss scale | 0.1 |
| World Model latent unimix | 1% |
| World Model free nats | 1 |
| Actor-Critic imagination horizon | 15 |
| Actor-Critic return lambda | 0.95 |
| Critic loss scale | 1 |
| Critic replay loss scale | 0.3 |
| Critic EMA regularizer | 1 |
| Critic EMA decay | 0.98 |
| Actor loss scale | 1 |
| Actor entropy regularizer | $3 \times 10^{-4}$ |
| Actor unimix | 1% |
| Actor RetNorm scale | $\text{Per}(R, 95) - \text{Per}(R, 5)$ |
| Actor RetNorm limit | 1 |
| Actor RetNorm decay | 0.99 |

