# OpenReview forum: "Boosting World Models Learning via Latent-Space Value Alignment"
_ICML.cc/2026/Conference — ICML 2026 regular_

### Official Review · Reviewer_hJRr · 2026-03-05

**Soundness:** 3
**Presentation:** 2
**Significance:** 2
**Originality:** 3
**Overall Recommendation:** 4
**Confidence:** 4

**Summary:**

This paper aims to combine maximum likelihood training of world models with value-aware training to capture task-relevant features important for policy learning. The authors propose a regularization of the latent space defined as the KL divergence between values predicted from posterior states and values predicted from prior states. Additionally, they introduce an adaptive weighting mechanism based on the dynamics loss of the model. The regularization is integrated into Dreamer and Storm and empirically evaluated.

**Compliance With Llm Reviewing Policy:**

Affirmed.

**Final Justification:**

The authors have well addressed my concerns including the incoherent motivation. I raised the score from 3 to 4.

**Key Questions For Authors:**

- Q1: How does the motivation of  including task relevant features which are important for downstream policy leanring connect to the KL regularizer compared to explicit value prediction?
- Q2: Why did you decide to use a KL regualizer? How does it compare to explicit value prediction e.g. similar to Dreamer v3 (2025 v2). What happens if you combine  explicit value prediction and the KL regualrizer?
- Q3: How does adding the regularizer affect the ELBO?

**Limitations:**

No limitations discussed.

**Strengths And Weaknesses:**

**Strengths**
- Strong results on the Atari 100k benchmark.
- Not tied to a single model. The approach generalizes to latent state world models in general, with Dreamer and Storm serving as examples.
- The latent-space alignment principle generalizes to reward and continue prediction.
- Little computational overhead.

**Weaknesses**
- The motivation is inconsistent with the implementation. Value-aware methods aim for a latent representation that captures task-relevant features important for downstream policy learning. However, the $L_{var}$​ regularizer does not shape the posterior to include features relevant for value prediction. It only aligns posterior and prior states to predict the same values. This distinction from conventional value-aware methods needs explicit discussion. It might also highlight a different phenomenon: the dynamics loss alone is insufficient to ensure task-relevant features are present in the prior, even when they are represented in the posterior. The results in Figure 3 support this intuition, as objects vanish without regularization, specifically in imagined trajectories that rely on the prior.
- There are no ablation studies on the form of the regularizer. It is unclear why this regularizer is preferable to explicitly predicting values.
- The motivation is somewhat circular. Figure 3.2 shows the result of the proposed method yet is used as motivation, which is problematic since motivation should be established independently of the method's outcomes. A separate example such as a toy example illustrating the difference between maximum likelihood training and value alignment is missing.
- Figure quality, size, and readability are poor.

---

> ### Author Rebuttal · Authors · 2026-03-31
>
> Dear Reviewer hJRr,
>
> We sincerely appreciate your constructive feedback！Below, we address your specific questions in detail.
>
> ---
>
> **W1&Q1: Motivation: KL value alignment vs. explicit value prediction.**
>
> **A1:** Good question! While explicit value prediction head shapes the posterior to include task-relevant features, it faces two key issues: (1) the posterior is connected to the reward head, the reconstruction head, and the continuation head. This inevitably subjects the posterior shaping to gradient dominance, where the scalar value-prediction gradients are easily overwhelmed by the high-dimensional image reconstruction loss; (2) even if the posterior contains task-relevant information, the dynamics representation loss treats all feature dimensions indiscriminately. Lacking a mechanism to prioritize task-critical information, the prior often fails to distill decision-relevant features from the posterior.
>
> In contract, our latent-space value alignment not only avoids the gradient competition among different prediction heads, but also prevents the prior from ignoring the task representations present in the posterior. Furthermore, while it does not explicitly reshape the posterior as a value prediction head would, our mechanism compels the prior and posterior to prioritize alignment in value-sensitive regions, thereby indirectly injecting task awareness into the representation learning process.
>
> Thank you for your constructive suggestion! We will emphasize this distinction in our revised manuscript.
>
> ---
>
> **W2&Q2: Why use KL regualizer? Comparison and combination with explicit value prediction (DreamerV3 2025 v2).**
>
> **A2:** Since DreamerV3 represents value outputs as parameterized probability distributions instead of scalars, we chose KL divergence to align the prior and posterior distributions. The following table presents an ablation study comparing this KL approach with the Mean Squared Error (MSE) of the scalar expected values.
>
> Given that the value outputs in DreamerV3 are parameterized as probability distributions rather than simple scalars, we employ KL divergence to measure the discrepancy between the prior and posterior value distributions. In the table below, we provide an ablation study comparing our KL-based formulation against the MSE calculated between the scalar expected values of the prior and posterior: $MSE(\mathbb{E}[V_{\theta}(v_t|s_t)], \mathbb{E}[V_{\theta}(\tilde{v}_t|\tilde{s}_t)])$.
>
> | Atari | Alien | Gopher | Krull | KungFuMaster |
> | --- | --- | --- | --- | --- |
> | DreamerV3 | 876 | 1684 | 8365 | 16375 |
> | MSE | 964 | 1956 | 8371 | 19460 |
> | KL | 1233 | 2807 | 9821 | 21075 |
>
> While MSE outperforms the baseline, it underperforms the KL formulation. By aligning the full distributional geometry and uncertainty, KL divergence provides richer, smoother gradient signals for the latent representation.
>
> To further distinguish KL value alignment from explicit prediction, we conducted experiments based on the DreamerV3 (2025 v2) framework across four distinct configurations: (a) baseline without value-aware components; (b) baseline + replay value loss; (c) baseline + var; and (d) baseline + both replay value loss and var.
>
> | Atari | Alien | Gopher | Krull | KungFuMaster |
> | --- | --- | --- | --- | --- |
> | (a) | 1049 | 2393 | 8040 | 25842 |
> | (b) | 942 | 2167 | 9243 | 26220 |
> | (c) | 1406 | 2726 | 9769 | 26564 |
> | (d) | 1194 | 2947 | 11303 | 27586 |
>
> The experimental results demonstrate that latent-space value alignment yields more robust and superior performance gains compared to replay value loss. Furthermore, combining these two mechanisms reveals a synergistic effect across the majority of tasks, leading to overall performance boost.
>
> ---
>
> **Q3: How does adding the regularizer affect the ELBO?**
>
> **A3:** We have visualized the ELBO for the Krull and KungFuMaster throughout the training process. The detailed visualizations are accessible via the following link: https://anonymous.4open.science/r/xxxx-3521/README.md. As demonstrated in the plots, the integration of our value-alignment regularization exerts a negligible impact on the ELBO, confirming that our method preserves the structural integrity of the variational objective.
>
> ---
>
> **W3: A separate example such as a toy example illustrating the difference between maximum likelihood training and value alignment is missing.**
>
> **A4:** To illustrate the distinction between maximum likelihood (ML) and value alignment, consider an autonomous driving scenario. Visual inputs contain both decision-critical features (e.g., traffic lights) and task-irrelevant noise (e.g., moving clouds). Because standard ML models optimize unweighted log-likelihood, large, high-entropy distractors inevitably dominate the reconstruction loss, causing the model to neglect vital targets. In contrast, value alignment compels the model to filter out irrelevant noise and focus its capacity strictly on decision-critical entities, enhancing representational efficiency.

---

> > ### Author Rebuttal · Reviewer_hJRr · 2026-04-03
> >
> > Thank you for the detailed rebuttal. My questions and concerns have been addressed.
> >
> > However, W1 & Q1 still require a substantial rewrite of the motivation. The cleaner narrative is: value prediction can enforce value awareness in the posterior, but fights reconstruction gradients. More importantly, even when the posterior contains value-relevant features, the dynamics loss does not guarantee to propagate them to the prior. KL value alignment directly addresses this gap and is orthogonal to value prediction, not a replacement. Prior-posterior value alignment alone appears more impactful than value prediction, and combining both gives best performance. But it also might fight state KL gradients.
> >
> > The prior-posterior value awareness gap is the novel and strongest insight of this paper and should be the central motivation.

---

> > > ### Author Response · Authors · 2026-04-05
> > >
> > > Dear Reviewer hJRr,
> > >
> > > Thank you so much for your constructive response and for providing such an incredibly insightful synthesis of our work. We are thrilled that our previous rebuttal successfully addressed your technical questions and concerns.
> > >
> > > We agree that the prior-posterior value awareness gap is the novel and strongest insight of this paper and should serve as the central motivation. Indeed, as illustrated in Figure 1 of our original manuscript, distinct from explicit value prediction, we reinforce task-relevant representations through prior-posterior value alignment in the latent space. We are very encouraged that you explicitly highlighted this aspect, as it aligns perfectly with the original and primary motivation of our work.
> > >
> > > Following your suggestions, we will organically integrate your proposed "cleaner narrative" into the revised manuscript. Specifically, we will center our revisions around this specific narrative:
> > >
> > > 1. **The Representation Bottleneck:** We will clarify that while explicit value prediction can enforce value awareness in the posterior, it inevitably fights high-dimensional reconstruction gradients.
> > >
> > > 2. **The Prior-Posterior Value Awareness Gap:** More importantly, we will highlight that even when the posterior successfully captures value-relevant features, the standard dynamics loss provides no guarantee that these task-critical features are propagated to the prior.
> > >
> > > 3. **Our Orthogonal Value Alignment Solution:** We will present our KL value alignment specifically as the mechanism designed to address this prior-posterior gap, explicitly stating that it is orthogonal to value prediction, not a replacement.
> > >
> > > 4. **Combined Efficacy:** We will incorporate the new ablation results from the rebuttal showing that prior-posterior value alignment alone is highly impactful, and that combining both mechanisms can also yield better performance.
> > >
> > > Your constructive feedback has undeniably elevated the theoretical framing and core message of our paper. Thank you once again for your time, engagement, and invaluable guidance to help us improve this work!
> > >
> > > Best regards,
> > >
> > > Authors of Submission 31833

---

### Official Review · Reviewer_saHJ · 2026-03-11

**Soundness:** 2
**Presentation:** 3
**Significance:** 3
**Originality:** 2
**Overall Recommendation:** 4
**Confidence:** 4

**Summary:**

The paper proposes a lightweight way to improve model-based RL by adding latent-space value alignment to standard world models.
Instead of matching a scalar value loss, it regularizes the model by minimizing the KL divergence between the value distributions induced by prior and posterior latent states, encouraging the representation to focus on decision-relevant features while preserving accurate dynamics.
An adaptive weighting scheme with a warm-up phase stabilizes training, and experiments on Atari 100k and DeepMind Control show consistent gains over DreamerV3 and STORM with little extra overhead.

**Compliance With Llm Reviewing Policy:**

Affirmed.

**Final Justification:**

I appreciate the authors' rebuttal which addressed my main concerns. I still think that my initial score is a fair assessment. Thus I keep my score.

**Key Questions For Authors:**

The following questions relate to the weaknesses above:
- Did you compare the adaptive weighting scheme against additional static coefficients besides $\beta_{var} = 0.5$?
- Was the 10k-step warm-up also applied to the static-weight baseline in Figure 5?
- In the "+ Reward & Continue Alignment" experiments, is value alignment also included, and what exact losses / coefficients are used?

**Limitations:**

yes

**Strengths And Weaknesses:**

Strengths:

- Conceptually simple and well-motivated idea.
- Lightweight modification that requires minimal changes to existing methods and has negligible computational overhead.
- Strong empirical performance on established benchmarks.

Weaknesses:

- The empirical support for the adaptive weighting mechanism is still somewhat inconclusive.
   The paper's main ablation compares the proposed adaptive scheme only against a single static baseline with $\beta_{var} = 0.5$, while the method itself combines both a 10k-step warm-up and an adaptive weight based on the dynamics loss.
   As a result, it is difficult to disentangle whether the gains come primarily from adaptivity, the weight scale, or the warm-up phase.
   A broader comparison against multiple fixed coefficients and a clearer statement of whether the static baseline also uses the warm-up would strengthen this part of the paper.
- The generalizability analysis would benefit from clearer experimental detail.
   In Section 4.5, the paper states that the alignment idea is extended beyond value to reward and continuation signals via KL-based alignment between prior and posterior distributions, but the exact loss definitions, coefficients, and training setup are not stated in the main text.
   In particular, it is hard to tell whether the "+ Reward & Continue Alignment" results are obtained in addition to value alignment or instead of it, and whether the same adaptive weighting is used in those experiments.
   This makes the scope of the claimed generality harder to assess.
- There are some presentation and notation issues that reduce clarity:
   - The transition into the robustness-to-noise experiment around Table 3 could be introduced more clearly.
   - Table 7 is labeled as an "ablation study on computational overhead," although it reads more like a resource comparison.
   - The critic/actor parameters appear in Appendix Eq. (6), while related symbols already appear earlier in the main text.
   - The world model is introduced as parameterized by $\alpha$, but Eqs. (3) and (5) use $\phi$.

The paper presents a simple and practically useful method with strong benchmark results, but the evidence for adaptive weighting and generalizability is not fully convincing, which lowers the soundness score.
Overall, the paper is slightly above the acceptance threshold.

---

> ### Author Rebuttal · Authors · 2026-03-31
>
> To Reviewer saHJ
>
> We sincerely appreciate your positive comments for the conceptual simplicity, lightweight nature, and strong empirical performance of our method.! In response to your questions, we have prepared the following detailed answers:
>
> ---
>
> **W1&Q1: Did you compare the adaptive weighting scheme against additional static coefficients besides 0.5?**
>
> **A1:** In our initial exploration phase, we extensively evaluated the sensitivity of the model to the alignment weight by testing various static coefficients. Specifically, we conducted an ablation study comparing five fixed weights: $\beta_{var} \in \[0.1, 0.3, 0.5, 1.0 \]$. The comparative results are summarized in the table below:
>
> | Atari | Alien | CrazyClimber | DemonAttack | BattleZone |
> | --- | --- | --- | --- | --- |
> | STORM | 1054 | 47473 | 195 | 7080 |
> | + static weight 0.1 | 1124 | 49465 | 198 | 7860 |
> | + static weight 0.3 | 1253 | 51785 | 191 | 7590 |
> | + static weight 0.5 | 988 | 55096 | 180 | 8480 |
> | + static weight 1.0 | 923 | 53260 | 167 | 9165 |
> | + adaptive weight | 1361 | 57335 | 205 | 10140 |
>
> Our experiments indicate that while small static weights yield modest improvements over the baseline, excessively large static weights impose over-prioritized constraints on value alignment that stifle dynamics representation learning, ultimately leading to a degradation in overall performance. By introducing the adaptive weighting mechanism, our method effectively navigates the trade-off between value alignment and dynamics losses, ensuring a consistent and stable boost in performance across diverse tasks. We will incorporate these additional results and analyses into the revised manuscript.
>
> ---
>
> **Q2: Was the 10k-step warm-up also applied to the static-weight baseline?**
>
> **A2:** Yes, to ensure a fair comparison and control for experimental variables, the 10k-step warm-up phase was consistently applied to the static-weight baseline as well. We will explicitly clarify this detail in the revised manuscript to avoid any ambiguity.
>
> ---
>
> **W2&Q3: In the "+ Reward & Continue Alignment" experiments, is value alignment also included, and what exact losses / coefficients are used?**
>
> **A3:** No, value alignment was not included. The results for '+ Reward & Continue Alignment' were obtained in place of value alignment to specifically demonstrate the generalizability of our proposed latent-space alignment mechanism. This confirms that any task-relevant, behaviorally grounded signal, not just the value function, can serve as an effective alignment anchor.
>
> For reward alignment, given that the reward predictor outputs a discrete probability distribution, we utilized KL divergence to minimize the discrepancy between the reward distributions predicted from posterior and prior states: $L_{rar} = KL[p_{\alpha}(r_t|s_t) \parallel p_{\alpha}(\tilde{r}_t|\tilde{s}_t)]$.
>
> Regarding the continuation flag, as the predictor outputs a scalar probability, we employed Mean Squared Error (MSE) to align the predicted continuation values from the prior and posterior states: $L_{car} = MSE[p_{\alpha}(c_t|s_t), p_{\alpha}(\tilde{c}_t|\tilde{s}_t)]$.
>
> Meanwhile, we maintained the adaptive weighting mechanism and the warm-up phase for these additional alignment objectives. We will explicitly specify these loss formulations and the corresponding experimental setup in the revised manuscript.
>
> ---
>
> **W3: Presentation and Notation Issues: (1) Transition to noise-robustness needs clarity. (2) Table 7 is more a resource comparison than ablation. (3) Critic/actor params appearing in Appendix but symbols used earlier. (4) Parameterization mismatch ( $\alpha$ vs. $\phi$).**
>
> **A4:** We sincerely thank the reviewer for your meticulous reading and constructive feedback. We believe these corrections will significantly enhance the clarity and rigor of our manuscript.
>
> Regarding (1): We will add a transitional paragraph at the beginning of Section 4.3 (around Table 3) to clearly explicitly motivate the noise-injection experiments: specifically, to directly evaluate the model's ability to filter out task-irrelevant distractors.
>
> Regarding (2): We agree. We will rename Table 7 to ‘Resource and Computational Overhead Comparison.'
>
> Regarding (3): We will formally define the actor and critic parameters in Section 3.2 of the main manuscript.
>
> Regarding (4): We will correct Equations 3 and 5 to consistently use $\alpha$ to denote the world model parameters.

---

> > ### Author Rebuttal · Reviewer_saHJ · 2026-04-02
> >
> > Thank you for the clarifications. The rebuttal addressed my main concerns, and I appreciate the additional explanations.
> >
> > Based on the response, I am more positive about the paper’s soundness. Overall, I believe the paper is solid and my concerns have been sufficiently addressed. My overall recommendation remains the same.

---

### Official Review · Reviewer_bryQ · 2026-03-12

**Soundness:** 3
**Presentation:** 3
**Significance:** 3
**Originality:** 3
**Overall Recommendation:** 5
**Confidence:** 3

**Summary:**

This paper presents a new type of world model architecture, more specifically a value-aligned world model. This design aims to bridge the two paradigms of maximum likelihood and value-aware world models. This allows the model to focus the learning on decision-critical dynamics and not get distracted by secondary details. The proposed method uses an adaptive weighting mechanism that balances reconstruction fidelity with task-awareness. The paper present experiments across 46 environments from the Atari 100k and DeepMind Control benchmarks. The results demonstrate significant performance increases.

**Compliance With Llm Reviewing Policy:**

Affirmed.

**Key Questions For Authors:**

1. Have you evaluated the proposed method on other SOTA world-model architectures?
2. How does the method perform in environments that require heavy exploration or can only be trained on poor quality data?
3. Could theoretically justify why this method of combining maximum likelihood and value-aware world models is more efficient than existing approaches? Could you provide some theoretical guarantees for your proposed method?

**Limitations:**

Yes.

**Strengths And Weaknesses:**

Strengths:
1. This is an important problem with an elegant solution. This is the first work that proposes an efficient solution for combining maximum likelihood and value-aware world models.
2. The adaptive weighting mechanism is an effective solution that makes the optimisation problem more stable. It balances the trade-off between reconstruction fidelity and value alignment.
3. The empirical results look promising. An extensive evaluation was run on the Atari and DeepMind Control Benchmark and strong performance results are demonstrated in Atari.

Weaknesses:
1. The improvements are less significant in DMC benchmark. The authors explain why, stating that DMC is not the ideal benchmark to test the proposed method. Hence, it would be interesting to get a clearer definition of what kind of environments this method was designed for and see more experiments on a benchmark that is more suitable to test the problem.

---

> ### Author Rebuttal · Authors · 2026-03-31
>
> To Reviewer bryQ
>
> We sincerely appreciate your positive comments! Below, we address your specific questions and constructive feedback in detail.
>
> ---
>
> **W1: Clarify target scenarios and add experiments on more suitable benchmarks.**
>
> **A1:** As discussed in Section 3.1, our approach primarily targets environments with constrained model capacity, high-dimensional inputs, and dense task-irrelevant distractors. In standard DeepMind Control (DMC) environments, observations are typically clean and low-redundancy. Consequently, standard maximum likelihood models already utilize their capacity efficiently, leaving limited room for improvement via value alignment. To further demonstrate our method's efficacy, we evaluated it on the vision-based Meta-World benchmark. These robotic manipulation tasks require the world model to precisely reconstruct both the arm's configuration and complex object interactions from high-dimensional visual inputs. The results are summarized in the table below.
>
> | Meta-World | box close | door open | lever pull |
> | --- | --- | --- | --- |
> | DreamerV3 | 1692 | 1993 | 1204 |
> | DreamerV3 + Var | 1909 | 2512 | 1670 |
>
> Experiments demonstrate that our algorithm enhances robotic manipulation, particularly in visually complex environments.
>
> ---
>
> **Q1: Evaluation on other SOTA architectures?**
>
> **A2:** While originally evaluated on DreamerV3 and STORM, we have now extended our method to two additional state-of-the-art architectures: TWISTER[1] and DIAMOND[2]. Given time and compute constraints, we focused these supplementary experiments on a representative subset of environments (Alien, Bank Heist, KungFuMaster, and BattleZone). Results are summarized below:
>
> | Atari | Alien | Bank Heist | KungFuMaster | BattleZone |
> | --- | --- | --- | --- | --- |
> | TWISTER | 954 | 893 | 22452 | 9984 |
> | TWISTER + Var | 1165 | 1082 | 24846 | 11640 |
>
> | Atari | Alien | KungFuMaster |
> | --- | --- | --- |
> | DIAMOND | 785 | 17685 |
> | DIAMOND + Var | 946 | 18958 |
>
> Results show that our value-alignment mechanism consistently improves performance across diverse architectures, ranging from Transformer-based (TWISTER) to diffusion-based (DIAMOND) world models.
>
> [1] Learning Transformer-based World Models with Contrastive Predictive Coding. 2025 ICLR
>
> [2] Diffusion for World Modeling: Visual Details Matter in Atari. 2024 NIPS
>
> ---
>
> **Q2: Performance in hard-exploration or low-quality data scenarios?**
>
> **A3:** We thank the reviewer for highlighting the critical challenges of hard-exploration and low-quality data regimes.
>
> **Hard-Exploration:** In tasks like PrivateEye and Frostbite, early-stage value signals are often flat. Prematurely imposing value-alignment constraints can suppress the representational diversity necessary for exploration, leading to the observed regressions. This issue can be effectively mitigated by extending the warm-up phase, delaying alignment until meaningful value gradients emerge. To evaluate this, we conducted an ablation study using DreamerV3 (2023 v1)  in the table below:
>
> | Atari | PrivateEye | Frostbite |
> | --- | --- | --- |
> | baseline | 2331 | 1628 |
> | 1e4 warm-up| -115 | 348 |
> | 3e4 warm-up| 542 | 943 |
> | 5e4 warm-up| 2841 | 1537 |
>
> As observed in the results, extending the warm-up phase leads to performance gains, even surpassing the baseline.
>
> **Low-Quality Data (Offline RL):** Our approach is naturally suited for offline settings, as filtering out background redundancy compels the world model to focus on task-relevant dynamics. Following the methodology in [3], we adapted DreamerV2 for the V-D4RL benchmark. Due to time and compute constraints, we provide preliminary evaluations on two representative environments. The results confirm that our method consistently enhances offline performance:
>
> | Env | Offline DV2 | Offline DV2 + Var |
> | --- | --- | --- |
> | walker-walk mixed | 56.5 | 60.4 |
> | cheetah-run mixed | 61.6 | 62.8 |
>
> We sincerely appreciate these valuable suggestions. Comprehensively addressing hard-exploration and low-quality data settings remains a highly promising direction for our future research.
>
> [3] Challenges and Opportunities in Offline Reinforcement Learning from Visual Observations. 2023 TMLR
>
> ---
>
> **Q3: Could theoretically justify why this method is more efficient than existing approaches?**
>
> **A4:** Efficiency Intuition: Explicit value-aware methods suffer from gradient dominance, as high-dimensional reconstruction gradients overwhelm scalar value gradients during joint optimization, causing instability. In contrast, our latent-space alignment circumvents this gradient competition and ensures the prior retains posterior task representations.
>
> Meanwhile, based on the Value Equivalence Principle [4], a world model only needs to be 'value-equivalent'  rather than perfectly reconstructing true transition dynamics. Our alignment adheres to this principle.
>
> [4] The Value Equivalence Principle for Model-Based Reinforcement Learning. 2020 NIPS

---

> > ### Author Rebuttal · Reviewer_bryQ · 2026-04-04
> >
> > The authors have addressed my concerns and I will keep my score.

---

### Official Review · Reviewer_RwzG · 2026-03-13

**Soundness:** 2
**Presentation:** 3
**Significance:** 2
**Originality:** 3
**Overall Recommendation:** 4
**Confidence:** 3

**Summary:**

This paper proposes a Value-Aligned World Model that aims to leverage the strengths of both (1) maximum-likelihood world models (stable dynamics modeling) and (2) value-aware world models (task-relevant representations), while avoiding explicit value-prediction heads or heavy external priors. The core idea is a latent-space value-alignment regularizer that aligns the distributional value predictions computed from posterior vs. prior latent states by minimizing a KL divergence between their value distributions. The authors evaluate the proposed approach by adding this regularizer to DreamerV3 (2023 v1) and STORM, reporting improved results on Atari-100k and DeepMind Control Suite (DMC).

**Compliance With Llm Reviewing Policy:**

Affirmed.

**Final Justification:**

The authors have adequately addressed my concerns. Therefore, I increase my score to 4.

**Key Questions For Authors:**

1. Can the proposed regularizer provide additional gains on top of already value-aware model backbones, such as DreamerV3 (2025 v2) with replay value loss?
2. How robust is the method in sparse-reward or hard-exploration setup, and are there possible ways to mitigate the regressions observed on games such as PrivateEye and Frostbite?
3. For newer models that already include a replay value loss (DreamerV3 v2), could you clarify what latent-space value alignment is intended to add beyond this mechanism? In particular, is it meant to complement replay value loss, or is it better understood as an alternative way of making the world model value-aware? If the two are complementary, it would be helpful to know whether results on top of DreamerV3 v2 would further support the practical value of the proposed method.

**Limitations:**

yes

**Strengths And Weaknesses:**

Strengths:
1. The proposed approach is simple and easy to implement. The regularizer is added to the world-model loss, and the paper also reports very small added computational overhead (Table 7).
2. The paper demonstrates solid empirical results on Atari-100k. Specifically, the method improves performance in 24 out of 26 games for both the DreamerV3 and STORM variants, suggesting that the gains are broad and consistent rather than being limited to only a few tasks.
3. The paper provides reasonable ablation and generalization checks. The adaptive-weight ablation in Table 6 supports the effectiveness of the proposed weighting mechanism over a static-weight baseline, and Table 5 shows that the same latent-space alignment idea can also be extended from value signals to reward and continuation signals, which strengthens the paper’s claim that the approach is not limited to value alignment alone.

Weaknesses:
1. The proposed approach appears less convincing in settings where the value signal is weak or noisy. As the authors noted, there is degraded performance on sparse-reward Atari games such as PrivateEye and Frostbite, and attributes them to unreliable value targets early in training. This suggests that the proposed regularization may trade off exploration for task-focus in hard exploration domains, but the paper does not provide a concrete mitigation for this failure mode.
2. Some of the mechanistic claims are supported by relatively narrow analysis. For example, the noise-robustness experiment (Table 3) is conducted on only two proprioceptive tasks, and the quantitative evidence for improved world modeling (Table 4) is reported only on KungFuMaster.
3. The practical significance relative to newer value-aware world models remains somewhat unclear. The paper intentionally builds on DreamerV3 (2023 v1) to isolate the effect of the proposed regularizer, while noting that DreamerV3 (2025 v2) already includes a value-aware replay value loss. Although this design choice is reasonable for controlled evaluation, it leaves open whether the proposed latent-space alignment provides complementary gains on top of already value-aware modern backbones, or whether its main benefit is primarily to equip older pure maximum-likelihood world models with a value-aware component.
4. The conceptual novelty appears to be a bit limited. The paper presents a lightweight and practical way to integrate value-aware signals into standard maximum-likelihood world-model training, but the main contribution is closer to a careful regularization design and integration strategy.

---

> ### Author Rebuttal · Authors · 2026-03-31
>
> To Reviewer RwzG,
>
> We sincerely appreciate your detailed feedback and constructive suggestions! Here is our response to your concerns:
>
> ---
>
> **W1&Q2: How robust is the method in sparse-reward or hard-exploration setup, and are there possible ways to mitigate the regressions observed on games such as PrivateEye and Frostbite?**
>
> **A1:** Good question! In sparse-reward tasks like PrivateEye and Frostbite, early-stage value signals are typically flat and uninformative. Prematurely imposing value-alignment constraints can suppress the representational diversity necessary for exploration, causing the observed regressions.
>
> A straightforward mitigation is extending the warm-up phase, delaying alignment until meaningful value gradients emerge. To evaluate this, we conducted an ablation study using DreamerV3, with results summarized below:
>
> | Atari | PrivateEye | Frostbite |
> | --- | --- | --- |
> | baseline | 2331 | 1628 |
> | 1e4 warm-up| -115 | 348 |
> | 3e4 warm-up| 542 | 943 |
> | 5e4 warm-up| 2841 | 1537 |
>
> As observed in the results, extending the warm-up phase leads to performance gains, even surpassing the baseline.
>
> Furthermore, integrating techniques like Random Network Distillation (RND) offers a more principled solution. By introducing intrinsic rewards, we can guide the world model to retain features for novel state discovery, balancing task-specific focus with exploratory breadth. Investigating sparse-reward robustness remains a promising avenue for our future research.
>
> ---
>
> **W2: Narrow analysis for noise-robustness experiment (Table 3) and quantitative evidence (Table 4).**
>
> **A2:** We have expanded the scope of our experiments in Table 3 and Table 4. The additional results and the derived conclusions remain consistent with our original findings.
>
> Regarding the noise-robustness experiments, we have incorporated new experimental results from the Walker Run environment:
>
> | Noise Ratio | DreamerV3 | DreamerV3+Var |
> | --- | --- | --- |
> | Insert 0% noise | 726 | 716 |
> | Insert 10% noise | 675 | 694 |
> | Insert 20% noise | 593 | 657 |
> | Insert 50% noise | 426 | 569 |
>
> Furthermore, for the quantitative evidence, we have provided additional quantitative results across the following environments:
>
> | CrazyClimber | MSE | Value error |
> | --- | --- | --- |
> | STORM | 0.00126 | 233.48 |
> | STORM + Var | 0.00119 | 64.67 |
>
> | Krull | MSE | Value error |
> | --- | --- | --- |
> | STORM | 0.00073 | 110.96 |
> | STORM + Var | 0.00059 | 16.44 |
>
> | Gopher | MSE | Value error |
> | --- | --- | --- |
> | STORM | 0.00040 | 32.40 |
> | STORM + Var | 0.00033 | 6.68 |
>
> | UpNDown | MSE | Value error |
> | --- | --- | --- |
> | STORM | 0.00447 | 80.69 |
> | STORM + Var | 0.00396 | 6.53 |
>
> ---
>
> **W3&Q1&Q3: Latent-space value alignment vs. replay value loss: Complementary or alternative? Can stacking them boost performance?**
>
> **A3:** Good question! While explicit value prediction head (e.g. replay value loss) shapes the posterior to include task-relevant features, it faces two key issues: (1) the posterior is connected to the reward head, the reconstruction head, and the continuation head. This inevitably subjects the posterior shaping to gradient dominance, where the scalar value-prediction gradients are easily overwhelmed by the high-dimensional image reconstruction loss; (2) even if the posterior contains task-relevant information, the dynamics representation loss treats all feature dimensions indiscriminately. Lacking a mechanism to prioritize task-critical information, the prior often fails to distill decision-relevant features from the posterior.
>
> In contract, our latent-space value alignment not only avoids the gradient competition among different prediction heads, but also prevents the prior from ignoring the task representations present in the posterior. Crucially, explicit value prediction and our latent-space alignment incorporate value awareness from distinct perspectives, rendering the two mechanisms entirely orthogonal rather than antagonistic.
>
> To further distinguish KL value alignment from explicit prediction, we conducted experiments based on the DreamerV3 (2025 v2) framework across four distinct configurations: (a) baseline without value-aware components; (b) baseline + replay value loss; (c) baseline + var; and (d) baseline + both replay value loss and var.
>
> | Atari | Alien | Gopher | Krull | KungFuMaster |
> | --- | --- | --- | --- | --- |
> | (a) | 1049 | 2393 | 8040 | 25842 |
> | (b) | 942 | 2167 | 9243 | 26220 |
> | (c) | 1406 | 2726 | 9769 | 26564 |
> | (d) | 1194 | 2947 | 11303 | 27586 |
>
> The experimental results demonstrate that latent-space value alignment yields more robust and superior performance gains compared to replay value loss. Furthermore, combining these two mechanisms reveals a synergistic effect across the majority of tasks, leading to a overall performance boost.

---

> > ### Author Rebuttal · Reviewer_RwzG · 2026-04-01
> >
> > Thank you for your reply. I believe the authors have adequately addressed my concerns and I will increase my score to 4 accordingly.

---

### Decision · Program_Chairs · 2026-04-30

**Decision:**

Accept (regular)

**Comment:**

The paper introduces a latent-space value alignment regularizer (L_var) for world models that uses KL divergence to align value predictions from prior and posterior states, together with an adaptive weighting mechanism driven by dynamics loss. The method is simple, principled, and imposes minimal computational overhead.

Reviewers were unanimously positive after the rebuttal. The paper's main strength is the breadth and rigor of its empirical validation. Beyond the original DreamerV3 and STORM experiments, the authors demonstrated during the rebuttal that L_var yields consistent gains on two additional architectures (TWISTER and DIAMOND), bringing the total to four world model families. The DreamerV3 v2 ablation was particularly informative: it showed that L_var is complementary to the replay value loss recently introduced in DV3 v2, ruling out the concern that the contribution would be subsumed by concurrent architectural advances. Ablations over the adaptive weighting coefficient confirmed that the proposed mechanism consistently outperforms fixed alternatives across environments.

Reviewer hJRr initially raised a valid concern about the motivation section conflating value-aware representation shaping with prior-posterior alignment. The authors acknowledged this distinction and provided a convincing technical argument about why KL alignment avoids gradient dominance from high-dimensional reconstruction losses. The reviewer was satisfied and raised their score. Minor notation inconsistencies flagged by Reviewer saHJ are acknowledged and straightforward to fix in the camera-ready.

The contribution is well-scoped, technically sound, and practically useful. The method generalizes across architectures with no environment-specific tuning, and the experimental evidence is thorough. I recommend acceptance. The authors should ensure the camera-ready version incorporates the promised motivation rewrite and notation fixes.